# Ciliary transition zone proteins coordinate ciliary protein composition and ectosome shedding

Liang Wang [1,7✉], Xin Wen[1,7], Zhengmao Wang [2,3,7], Zaisheng Lin[4], Chunhong Li[1], Huilin Zhou[1], Huimin Yu[1], Yuhan Li[1], Yifei Cheng[1], Yuling Chen[5], Geer Lou[6], Junmin Pan [2,3] & Muqing Cao [4✉]

The transition zone (TZ) of the cilium/flagellum serves as a diffusion barrier that controls the entry/exit of ciliary proteins. Mutations of the TZ proteins disrupt barrier function and lead to multiple human diseases. However, the systematic regulation of ciliary composition and signaling-related processes by different TZ proteins is not completely understood. Here, we reveal that loss of TCTN1 in *Chlamydomonas reinhardtii* disrupts the assembly of wedge-shaped structures in the TZ. Proteomic analysis of cilia from WT and three TZ mutants, *tctn1*, *cep290*, and *nphp4*, shows a unique role of each TZ subunit in the regulation of ciliary composition, explaining the phenotypic diversity of different TZ mutants. Interestingly, we find that defects in the TZ impair the formation and biological activity of ciliary ectosomes. Collectively, our findings provide systematic insights into the regulation of ciliary composition by TZ proteins and reveal a link between the TZ and ciliary ectosomes.

[1] School of Life Sciences, Jiangsu Normal University, 221116 Xuzhou, China. [2] Laboratory for Marine Biology and Biotechnology, Qingdao National Laboratory for Marine Science and Technology, 266071 Qingdao, China. [3] MOE Key Laboratory of Protein Sciences, School of Life Sciences, Tsinghua University, 100084 Beijing, China. [4] Key Laboratory of Cell Differentiation and Apoptosis of Chinese Ministry of Education, Department of Pathophysiology, Shanghai Jiao Tong University School of Medicine, 200025 Shanghai, China. [5] School of Life Sciences, Tsinghua University, 100084 Beijing, China. [6] Shanghai Biotree Biotech Co. Ltd, 201815 Shanghai, China. [7] These authors contributed equally: Liang Wang, Xin Wen, Zhengmao Wang. ✉email: wangliang@jsnu.edu.cn; muqingcao@sjtu.edu.cn

Cilia and flagella, conserved antenna-like structures protruding from cell surfaces, harbor a number of signaling-associated molecules that maintain cellular homeostasis and regulate tissue development[1–3]. To control the specific ciliary composition, a specialized zone at the base of cilia named the transition zone (TZ) serves as a gatekeeper to sort proteins into/out of the cilia[4–6]. Defects in TZ genes cause a variety of syndromes, termed ciliopathies, including Meckel–Gruber syndrome (MKS), nephronophthisis (NPHP), Joubert syndrome (JBTS), Senior–Loken syndrome (SLSN), and oral–facial–digital (OFD) syndrome, characterized by polydactyly, kidney cysts, ataxia, hyperpnea, nervous system degeneration, and developmental delay[7–12]. The function of the TZ structure relies on the integrity of two major complexes located at the TZ region, termed the MKS complex and NPHP complex, including MKS1, TMEM67, TMEM216, B9D1, B9D2, CEP290, and TCTN1-3 for the MKS complex and NPHP1, NPHP4, NPHP8, and NPHP2 for the NPHP complex[9,13]. Defects in any gene encoding a protein in the two complexes have been associated with various human diseases, indicating the indispensable function and different role of each member. To date, how each TZ protein uniquely and systematically controls the sorting of ciliary proteins for ciliary assembly and signaling remains largely unknown.

Ectosomes, vesicles secreted from the cell surface, are composed of various types of proteins and RNAs and mediate cellular communication in cell populations[14–18]. Surprisingly, the ciliary membrane is capable of generating ectosomes, and the shedding of ciliary ectosomes plays multiple roles in biological processes[19–25]. Unlike mammalian cells shedding off both ectosomes and exosomes from plasma membrane with distinct assembly mechanisms and diameters, cilia are the unique source of ectosomes derived from *C. reinhardtii* cells surrounded by cell walls, making *C. reinhardtii* as an ideal model for the study on the mechanism of ectosomes shedding[17,19,20]. However, whether the TZ, a major regulator of ciliary components and signaling, has functions in the release of ciliary ectosomes is a potentially interesting and important area of investigation.

In this work, we identify a mutant cell line of *C. reinhardtii*. The disrupted gene encodes a homolog of mammalian TCTN proteins; thus, we name the mutant *tctn1*. Our results show that *C. reinhardtii* TCTN1 localizes to the TZ region independent of CEP290 and NPHP4, and the loss of TCTN1 largely attenuates the formation of wedge-shaped structures in the TZ and glycocalyx on the surface of ciliary membrane. Taking advantage of cilia isolation in *C. reinhardtii*, we purify cilia from WT and three TZ mutants, *tctn1*, *nphp4*, and *cep290*. Proteomic analysis of the cilia indicates that each gate molecule plays a unique role in the control of ciliary proteins into/out of cilia, although some common functions are observed. Furthermore, we find that disruption of the TZ alters the size and biological activities of ciliary ectosomes. Systematic proteomics profiling and ectosome analysis show that each TZ protein has a unique function in ciliary events, including ciliary protein sorting and ciliary ectosome shedding.

## Results

**Loss of TCTN1 attenuates ciliogenesis in *C. reinhardtii*.** To identify new genes involved in ciliary assembly, we performed an unbiased forward genetic screen in a mutant library generated by insertional mutagenesis and found one cilia-deficient cell line. Restriction enzyme site-directed PCR (RESDA-PCR) showed that the insertion was located in the fifth exon of the *Cre03.g181450* gene (Fig. 1a), which encodes a 679 aa homologous protein of mammalian TCTN1, 2, and 3 (Fig. 1b). Since there is only one TCTN homolog in *C. reinhardtii*, we named the mutant *tctn1*. In contrast to wild-type cells with two equal-length cilia, *tctn1* cells

were palmelloid without visible cilia (Fig. 1c, d), indicating fail of lysis of mother cell wall after mitosis. Transformation of the *tctn1* mutant with an HA-tagged full-length *TCTN1* DNA fragment restored *TCTN1* expression (Supplementary Fig. 1a) and rescued the defect in ciliary assembly (Fig. 1c and Supplementary Fig. 1b), which confirmed that the ciliary phenotype was caused by the mutation in *TCTN1*. Autolysin, the proteolytic enzyme, is often used for lysis of cell wall to hatch daughter cells[19,26,27]. Upon autolysin enzyme treatment, *tctn1* cells hatched from the mother cell wall and assembled cilia (Fig. 1e). However, the average final ciliary length after hatching reached ~8 μm, which was shorter than that of WT cells (Fig. 1f, g), and occasionally bulges at the tip of short cilia were observed (Fig. 1e). It was reported that the TZ functioned in deciliation, ciliary assembly, and disassembly in other organisms[28–30], so we determined the excision, regeneration, and resorption of the cilia in the *tctn1* cells. Consistent with the WT and rescued cells, *tctn1* cells showed no defects in the deciliation process and completed the ciliary regeneration to reach the original length 2 h after deciliation by pH shock, but the mutants showed slower ciliary assembly kinetics (Fig. 1g). Interestingly, in the cilia shortening process induced by Nappi, *tctn1* cells showed faster ciliary disassembly kinetics (~5.864 μm/h) than those in WT (~3.363 μm/h) and rescued cells (~3.548 μm/h) (Fig. 1h).

TCTN1, 2, and 3 localize at the TZ in other model organisms[31,32]. Immunostaining analysis of the rescued cell line expressing TCTN1-HA showed that TCTN1 in *C. reinhardtii* is also a TZ protein at the base of the cilium (Fig. 1i). Consistently, polyclonal antibodies against endogenous *C. reinhardtii* TCTN1 showed the same TZ staining pattern in the WT and rescued lines but not in the *tctn1* mutant (Supplementary Fig. 1c). The polyclonal antibodies against endogenous TCTN1 showed additional signal in the cell body compared with the HA antibody (Fig. 1i and Supplementary Fig. 1c). Because the antibodies against the endogenous TCTN1 showed signals in cell bodies of the *tctn1* cells, which lost TCTN1 protein, it was very possible that the staining of the cell body was nonspecific. Consistent with previous studies for other TZ protein, TCTN1 was also associated with the ciliary base after deciliation and reciliation (Supplementary Fig. 1d)[33,34]. Immunoblot analysis of whole cells, cell bodies, and cilia verified that TCTN1 was not present in cilia (Fig. 1j), which is consistent with proteomic studies of cilia and the TZ in *C. reinhardtii*[35,36]. Taken together, these data demonstrated that *C. reinhardtii* TCTN1 is a TZ protein and functions in ciliogenesis.

**Loss of TCTN1 causes ultrastructural defects of cilia.** *C. reinhardtii* is an ideal model for transmission electron microscopy (TEM) studies of the ultrastructure of cilia, especially the TZ[37–39]. TEM revealed that the bulges (Fig. 1e) in the short cilia of *tctn1* cells were filled with electron-dense material (Fig. 2a). In the longitudinal sections through the TZ, the wedge-shaped structures that bridge the central "H" structure to the ciliary membrane were prominent in WT and rescued cells (Fig. 2b). However, the wedge-shaped structures were malformed, with some electron-dense material misplaced on the microtubule doublets (Fig. 2b). In cross sections, no evident defect in the Y-links that bridge the TZ microtubules and ciliary membrane was observed in *tctn1* (Fig. 2c). Meanwhile, quantification of the presence of wedge-shaped structures or Y-links was performed (Fig. 2c). Disruption of wedge-shaped connectors or Y-links impairs the connection between the microtubule axoneme and the ciliary membrane[34]. We measured the distance between the "H" structure and the ciliary membrane in WT, *tctn1*, and rescued cells and found that the distance was greater in *tctn1* mutant cells than that in WT or rescued cells (Fig. 2d). Interestingly, the

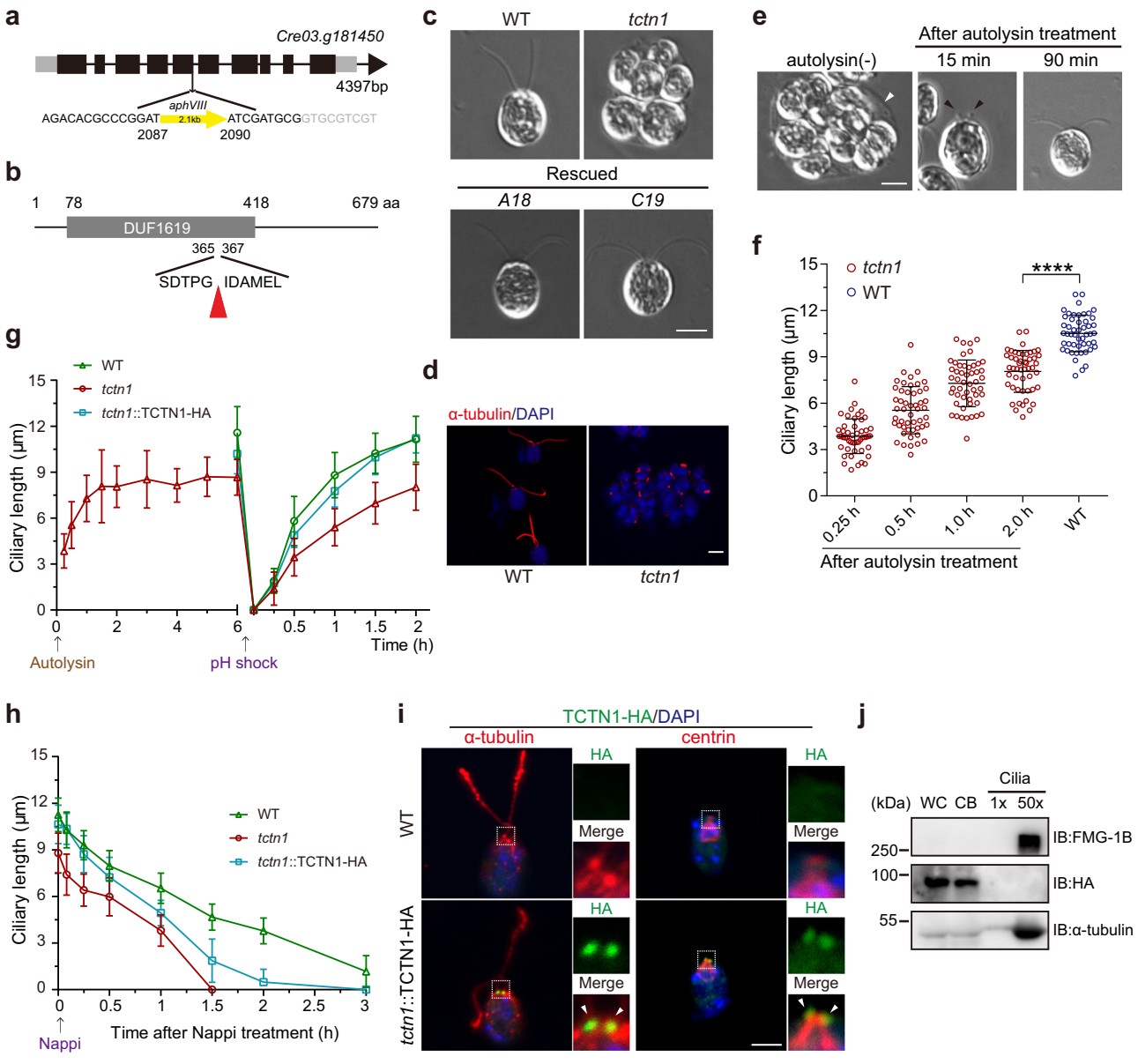

**Fig. 1 Characterization of the *C. reinhardtii tctn1* mutant. a** Diagram of the gene structure of *TCTN1* (*Cre03.g181450*) with the *aphVIII* DNA insertional site. **b** Diagram of the domain structure of the TCTN1 protein. The TCTN1 protein (679 aa) contains the DUF1619 domain (78–418 aa) with unknown function. The red arrowhead marks the insertional site. **c** DIC images showing the ciliary phenotypes of WT, mutant, and rescued cells (*A18, C19*). Scale bar, 5 μm. **d** Immunostaining images depicting the very short cilium within the palmelloid of *tctn1*. The red signals (by anti-α-tubulin antibody) mark the cilia and the blue signals (by DAPI) mark the nucleus. Scale bar, 5 μm. **e** DIC images depicting the ciliary phenotype of *tctn1* after treatment with autolysin. The mother cell walls are indicated by white arrowhead and the ciliary bugles are indicated by black arrowheads. Scale bar, 5 μm. **f** Scatter plot showing the elongation of cilia in **e**. Statistical significance was determined with an unpaired *t* test. ****P < 0.0001 by two tailed. **g** *tctn1* exhibited shorter cilia after hatching with autolysin and slower kinetics of ciliary assembly. The arrow indicates the time point of autolysin or pH shock treatment. **h** *tctn1* exhibited faster kinetics of ciliary disassembly. The arrow indicates the time point of Nappi treatment to induce ciliary shortening. **i** Immunostaining showing the localization of TCTN1 in the TZ. Cells were immunostained with HA (green) and α-tubulin (red, left) or centrin (red, right) antibodies. The nucleus was stained with DAPI (blue). The arrowheads indicate the TZ. The dotted boxes indicate the regions of higher magnification views. Scale bar, 5 μm. **j** Immunoblot analysis of the localization of TCTN1-HA in the rescued cell line. 1× (cilia) represents an equal proportion of cilia to that of the cell body (two cilia per cell body). 50× (cilia) represents equal cilia and cell body proteins. WC whole cell, CB cell body. Ciliary lengths are shown as the mean ± SD of 50 cilia. Source data are provided as a Source Data file.

glycocalyx on the surface of the ciliary membrane was largely disrupted in *tctn1* mutant cells, but the 9 + 2 microtubule axoneme was normal (Fig. 2e, f).

**Localization of TCTN1 is independent of CEP290 or NPHP4.** To further illustrate the localization of TCTN1, we performed immunostaining with antibodies against HA, acetylated α-tubulin, and CEP290 or NPHP4 and found that TCTN1 was distal to CEP290 but proximal to NPHP4 (Fig. 3a–d). Super-resolution microscopy confirmed localization of TCTN1, CEP290, and NPHP4 (Supplementary Fig. 2a). Immunogold TEM was used to localize TCTN1 more precisely within the TZ using the cells expressing TCTN1-HA with anti-HA antibodies. Though the

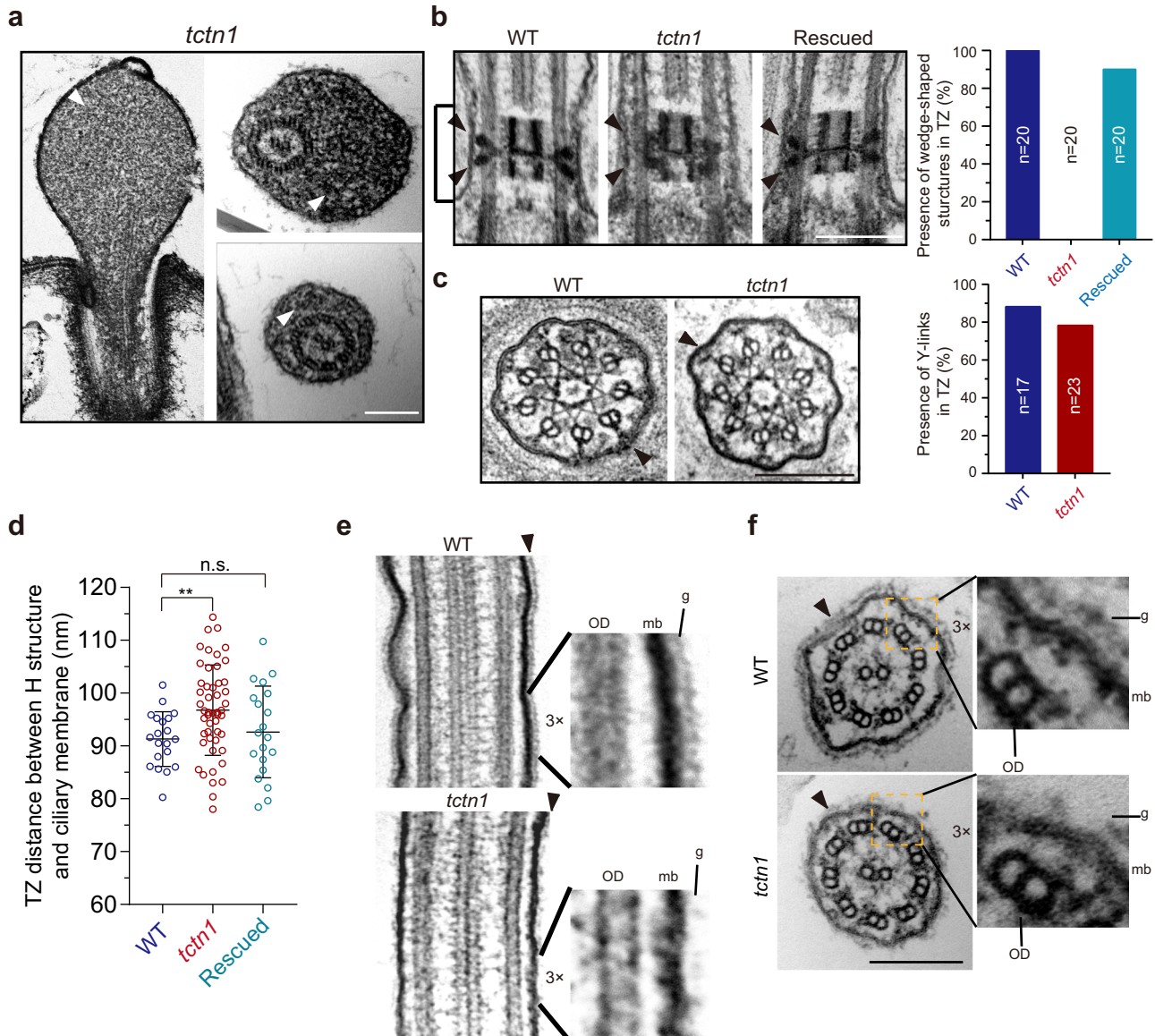

**Fig. 2 Loss of TCTN1 causes ultrastructural defects of the ciliary membrane and the TZ. a** EM images showing ciliary bulges with electron-dense material (white arrowheads) in a portion of *tctn1* cells. Scale bar, 200 nm. **b, c** Longitudinal sections (**b**) and cross sections (**c**) through the TZ (**b**, brackets) of WT, *tctn1* and rescued cells. The wedge-shaped structures (**b**, black arrowheads) and Y-links (**c**, black arrowheads) were indicated. Quantification of the presence of wedge-shaped structures in the longitudinal section ($n = 20$, number of wedge-shaped structures) and Y-links in the cross section ($n = 17$ for WT, $n = 23$ for *tctn1*, number represents the thin sections counted.). The presence of at least one Y-link in the thin section is considered as normal. Scale bar, 200 nm. **d** Scatter plot depicting the distances between the "H" structure and the ciliary membrane in WT, *tctn1*, and rescued cells. Data are the mean ± SD ($n = 20$). Statistical significance was determined with an unpaired *t* test. n.s., not significant; **$P < 0.01$ by two tailed. **e, f** TEM images of longitudinal (**e**) and cross (**f**) sections of the cilia from WT and *tctn1* cells. Glycocalyx on the surface of ciliary membrane was indicated with black arrowheads. The high magnification regions were indicated by dotted boxes. OD outer doublet microtubule, mb ciliary membrane, g glycocalyx. Scale bar, 200 nm. Source data are provided as a Source Data file.

ultra-structures were not well preserved, we observed that most gold particles were located at the transition zone periphery and associated with remnants of the transition zone membrane (Fig. 3e). Regarding the localization of CEP290 and NPHP4 revealed by previous studies[34,40], the localization of TCTN1 and these two TZ proteins were illustrated (Fig. 3f). Furthermore, we carried out immunostaining assays in *tctn1*, *cep290*, *nphp4*, and WT cells and found that the localization of TCTN1, CEP290, and NPHP4 was independent of the other two proteins (Fig. 3g).

**The phenotypes of *tctn1*, *cep290*, and *nphp4* are variable**. The integrity of the TZ affects ciliary morphology and motility. *cep290*

are often palmelloid and immotile, while *nphp4* have normal cilia with slightly defective swimming linearity[34,40]. We compared the phenotypes of *tctn1*, *cep290*, and *nphp4*. The *tctn1* cells were palmelloid and immotile in TAP medium, similar to *cep290* (Supplementary Fig. 3a). After autolysin treatment, *tctn1* cells were released from the mother cell wall, and the cilia were gradually assembled. ~90% of *tctn1* cells had cilia 3 h after autolysin treatment, and the final average ciliary length was 7.94 ± 1.14 μm (Supplementary Fig. 3a, b). For *cep290*, ~54% of cells had cilia with a final average length of 9.47 ± 0.54 μm (Supplementary Fig. 3a, b). Occasionally, we found that when the mutant *tctn1* cells were cultured in M medium (minimal media) at low cell

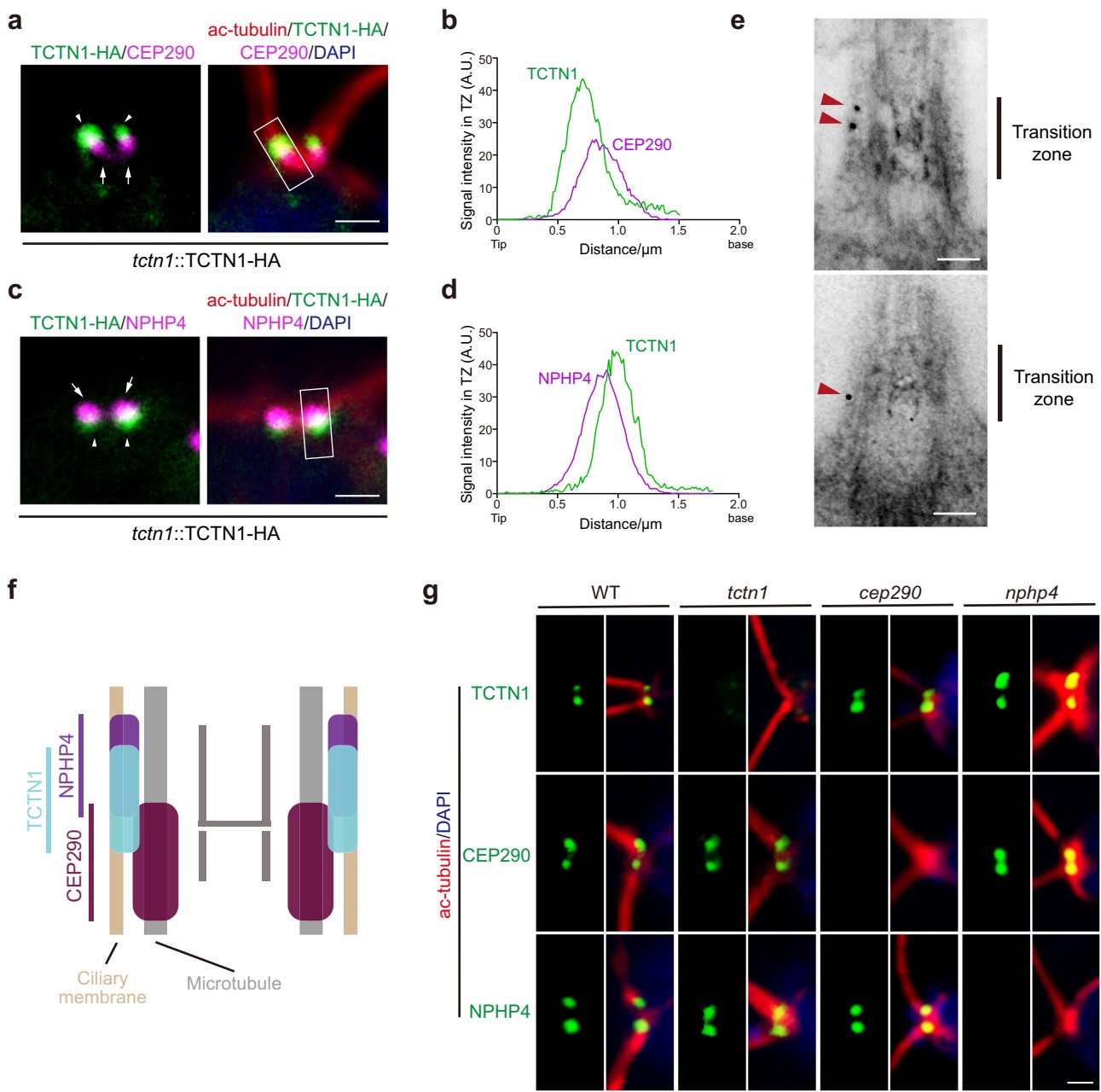

**Fig. 3 TCTN1 is located between CEP290 and NPHP4 independently. a–d** Immunostaining images and graphs depicting the colocalization of TCTN1 and NPHP4/CEP290. The rescued cells expressing TCTN1-HA were immunostained with anti-acetylated α-tubulin (ac-tubulin, red), anti-HA (green), and anti-CEP290 (**a**, magenta) or anti-NPHP4 (**c**, magenta) antibodies. The nucleus was stained with DAPI (blue). The arrowheads and arrows indicate the localization of TCTN1-HA and CEP290 (**a**)/NPHP4 (**c**), respectively. The scan plots of the rectangular gray value in the merged image show the relative intensities (arbitrary units, A.U.) of the indicated proteins in the TZ. Scale bar, 1 μm. **e** TEM imaging showing the longitudinal sections of TCTN1–HA transition zones labeled with anti-HA (10-nm gold, red arrowheads). Scale bar, 0.1 μm. **f** Schematic of the localization of TCTN1, CEP290, and NPHP4 at the TZ. **g** Immunostaining depicting the TZ localization of TCTN1, NPHP4, and CEP290 in WT, *tctn1*, *cep290*, and *nphp4* cells. Cells as indicated were immunostained with anti-acetylated α-tubulin (ac-tubulin, red), anti-TCTN1/CEP290/NPHP4 (green) antibodies. The nucleus was stained with DAPI (blue). Scale bar, 1 μm.

density, the *tctn1* cells were not palmelloid, and ~66% of cells had cilia (8.81 ± 0.75 μm) and swam normally, while this phenotype was not found in *cep290* (Supplementary Fig. 3a). Compared to WT cells (ciliary length, 11.42 ± 0.94 μm) and *nphp4* cells (ciliary length, 11.60 ± 0.98 μm), *tctn1* and *cep290* cells were paralyzed. Consistently, more ciliary bulges were observed in *tctn1* and *cep290* cells (Supplementary Fig. 3c). In addition, *tctn1* and *cep290* cells showed slower ciliary regeneration kinetics than WT and *nphp4* cells (Supplementary Fig. 3d). Therefore, these

collective data suggest that TCTN1 and CEP290 might play more essential roles in organization of the TZ and sorting of ciliary proteins than NPHP4.

**Loss of *TCTN1* systematically disrupts ciliary components**. It has been reported that TCTN proteins contribute to the concentration of several ciliary membrane proteins in mammalian cells and worms[32,41,42]. However, the systematic changes in the ciliary protein composition in TZ mutants are unknown. The cilia

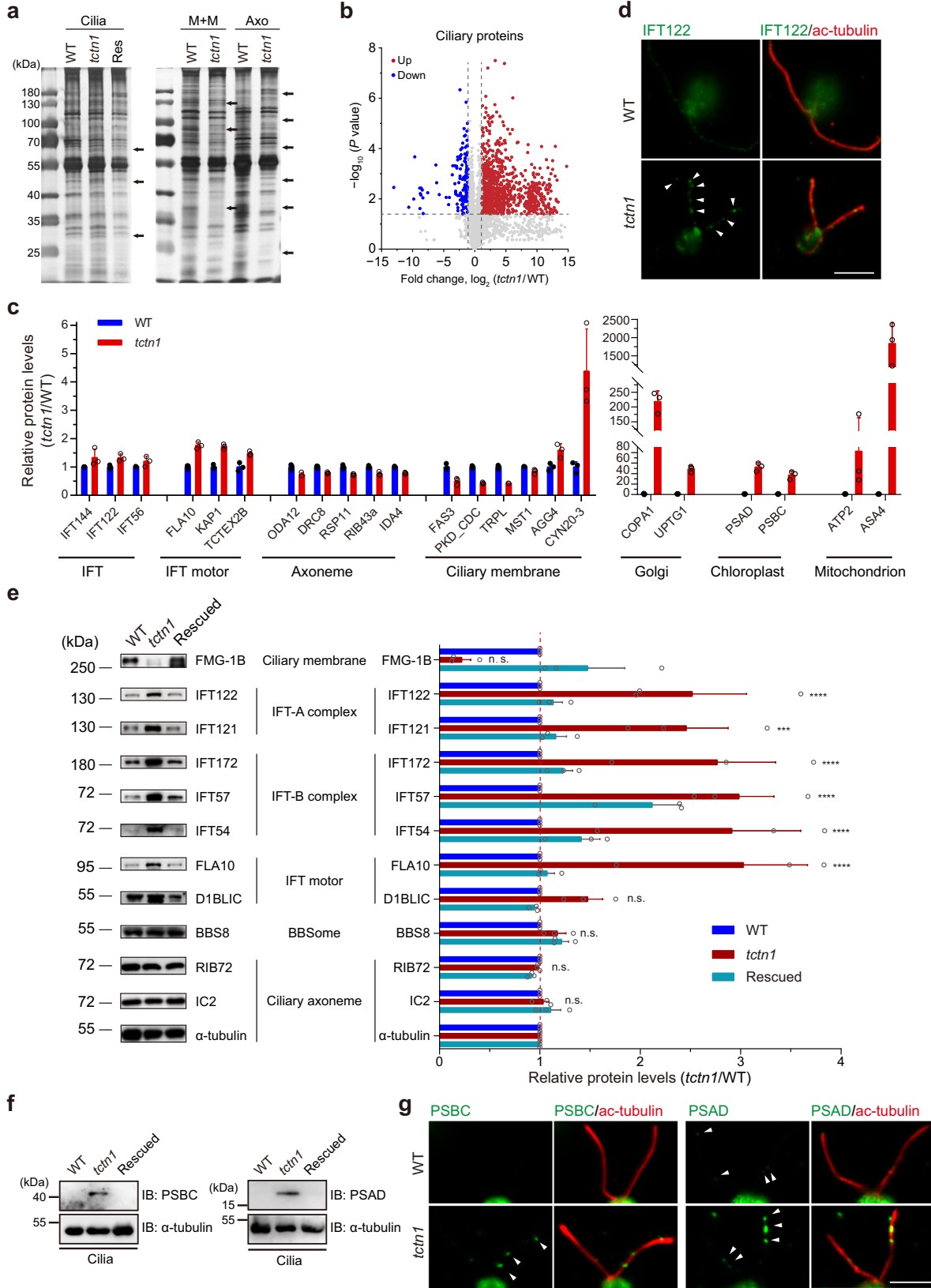

of WT and *tctn1* were isolated and subjected to silver gel staining and proteomic analysis. Total ciliary protein (Cilia) and the membrane-matrix (M + M) and axonemal (Axo) fractions from the WT, *tctn1* and/or rescued cells were analyzed via SDS–PAGE. Significant differences between the WT and *tctn1* cilia were identified (Fig. 4a). To profile the systematic changes in ciliary

proteins, we performed proteomic analysis of WT and *tctn1* cilia. Unexpectedly, a volcano plot showed that more proteins in cilia were increased in *tctn1* (Fig. 4b), which indicates that defects in the TZ not only impaired ciliary membrane protein concentration but also led to loss of the filter function to exclude other proteins (Fig. 4c). Clearly, the histogram showed an increase in IFT

**Fig. 4 Loss of TCTN1 attenuates the ciliary gating role of the TZ. a** SDS–PAGE and silver staining of isolated whole cilia or fractionations of cilia from WT, *tctn1*, and rescued cells expressing TCTN1-HA. Arrows indicate the differences among samples. Res rescued cells, Cilia ciliary samples. M + M, membrane and matrix samples of cilia. Axo axonemal samples of cilia. **b** Volcano plot displaying the differentially expressed proteins in cilia. The red and blue dots represent the upregulated and downregulated proteins in cilia of the *tctn1* mutant. The nonaxial vertical lines indicate fold change of *tctn1*/WT < 0.5 and >2. The nonaxial horizontal line represents *P* value of 0.05. Statistical significance was two-sided without adjustments. **c** Bar–scatter graph depicting the fold changes in the representative proteins in cilia between WT and *tctn1* in **b**. Data are the mean ± SD ($n = 3$). **d** Immunostaining images displaying the enrichment of IFT122 particles in the cilium of *tctn1*. Cells were immunostained with anti-IFT122 (green) and anti-acetylated α-tubulin (ac-tubulin, red) antibodies. The arrowheads indicate the accumulations of IFT122 in the cilium. Scale bar, 5 μm. **e** Immunoblot and statistical analysis of the isolated cilia from WT, *tctn1*, and rescued cells. α-tubulin was used as a loading control. The graph showing the gray value of the immunoreactive bands was prepared using the mean ± SEM ($n = 3$). Statistical significance to WT group was determined using two-way ANOVA with Dunnett's test. n.s., not significant. ***$P < 0.001$; ****$P < 0.0001$. **f** Immunoblot of cilia isolated from WT, *tctn1*, and rescued cells. The antibodies against PSBC and PSAD were used for the confirmation of proteomics results from **b**. **g** Immunostaining images showing enrichment of PSBC, PSAD in *tctn1*. Cells were immunostained with anti-PSBC (green, left), or anti-PSAD (green, right) and anti-acetylated α-tubulin (ac-tubulin, red) antibodies. The arrowheads indicate the accumulation of photosystem proteins in the cilium. Scale bar, 5 μm. Source data are provided as a Source Data file.

complexes, IFT motors, Golgi, chloroplast, and mitochondrion proteins in cilia (Fig. 4c). Immunofluorescence and immunoblot analyses confirmed our proteomic data (Fig. 4d, e and Supplementary Fig. 4). Surprisingly, the organelle proteins that presumably are not normally present in cilia but reside in the Golgi, chloroplast, and mitochondrion were transported into the cilia of the *tctn1* mutant (Fig. 4c). We verified these data via immunofluorescence and immunoblotting assays with antibodies against PSBC and PSAD (Fig. 4f, g). Immunofluorescence with anti-PSBC and anti-PSAD antibodies showed cup-shaped chloroplasts in cell bodies (Supplementary Fig. 5a, b) and abnormal accumulation of PSBC and PSAD in the cilia of *tctn1* cells (Fig. 4g). Collectively, the above results indicate that the defective TZ permitted entry of cytoplasmic proteins into cilia, which was consistent with the presumed function of the TZ in excluding nonciliary proteins and the gating roles of ciliary proteins in cilia.

**Each TZ protein specifically regulates ciliary composition.** The phenotypic differences of the TZ mutants suggest that each TZ protein plays specific roles in control of ciliary composition. To systematically assess the functions of the TZ proteins in ciliary composition regulation, we performed proteomic analysis of cilia purified from WT, *tctn1*, *cep290*, and *nphp4* cells. In total, 2845 proteins were identified, and their relative abundances were determined. A heatmap indicated that the protein profile of each mutant cell was significantly unique (Supplementary Fig. 6a). Although the heatmap pattern of ciliary proteins in *nphp4* was similar to that in the WT cilia (Supplementary Fig. 6a), ~30.9% of the proteins showed threefold changes in expression (>3 or <1/3, *nphp4*/WT) compared to the WT cilia. Mutations in *TCTN1* or *CEP290* seemed to cause more severe defects in the TZ, which led to 51.8% or 60.9% of proteins showing threefold changes (*tctn1*/WT or *cep290*/WT), respectively. Furthermore, according to the protein function, we performed grouped comparisons with WT and TZ mutants (Fig. 5a–d). Cargo transport-related proteins, including IFT/motor/BBS proteins, accumulated in the cilia of *tctn1* and *cep290* to different degrees (Fig. 5a), while ciliary structural proteins, such as axonemal components, showed various levels of downregulation in the cilia of *tctn1* and *cep290* (Fig. 5b). The regulation of ciliary membrane proteins seemed to be completely disrupted in the cilia of TZ mutants (Fig. 5c). Consistent with the proteomic results for *tctn1*, the nonciliary proteins from the Golgi, chloroplast, and mitochondrion were mistransported into the cilia of *cep290* mutants (Fig. 5d). The amount of ciliary membrane proteins, IFT proteins, and axonemal proteins was confirmed by immunoblot analysis (Supplementary Fig. 6b). Through immunofluorescence, we also verified that the chloroplast proteins of PSBC and PSAD were mislocated in the cilia of *cep290* and *tctn1* cells (Fig. 5e–g). These results

demonstrate that loss of each TZ protein leads to unique and systematic changes in ciliary composition, which account for specific disruptions in ciliary signaling.

**TZ proteins regulate the formation of ciliary ectosomes.** Cilia-derived ectosomes contain biologically active molecules and have multiple functions in the regulation of ciliary signaling[19,21,23,24,43,44]. From the ciliary proteomics data, we noticed that the amount of ectosome-related proteins was disrupted in the cilia of mutant cells (e.g., FOX1, AGG3, FEA2, AGG4, FAP212, CYN20-1) (Fig. 6a)[20]. Considering the essential role of the TZ in ciliary membrane composition control, it is possible that ciliary ectosome formation may be affected by defects in the TZ. Additionally, defects of the TZ mutant in hatching from the mother cell wall also suggested alteration of ectosome formation in the TZ mutant cilia.

In the mating process, gametic cells release large amounts of ectosomes that activate other gametes by binding to the cilia of cells with opposite mating types[22,25]. We mixed gametic cells (21*gr* × 6145c [WT]; *tctn1* × 6145c [*tctn1*]; *cep290* × 6145c [*cep290*]; *nphp4* × 6145c [*nphp4*]) at a ratio of 5:1, in which ectosomes from plus gametes constituted the majority. We purified ciliary ectosomes, followed by the negative staining TEM to visualize the morphology of ciliary ectosomes. Images revealed that the diameters of the ectosomes in the TZ-mutant groups were significantly smaller than those in the control group (Fig. 6b, c). The average diameter of ectosomes from WT was 198.20 ± 4.66 nm ($n = 500$), while the diameters of the ectosomes from *tctn1*, *cep290*, and *nphp4* were 87.35 ± 1.98 nm ($n = 500$), 83.99 ± 1.72 nm ($n = 500$), and 134.40 ± 3.93 nm ($n = 500$), respectively (Fig. 6c). Moreover, the size distribution varied greatly among these ectosomes. For *tctn1* and *cep290*, the diameter of most ectosomes was 50–100 nm, while the diameter of most ectosomes of *nphp4* was slightly larger, with a distribution range of 50–100 and 100–150 nm (Fig. 6d). These data indicate that the TZ structure plays a role in the formation of ciliary ectosomes.

**TZ regulates the composition and activity of ectosomes.** Our previous studies showed that ciliary ectosomes exhibit biological activity to induce gamete agglutination of single mating type gametes and activate cilium-generated signaling during the mating process[22]. This behavior of single mating type gametes agglutination induced by the ciliary ectosomes is very similar to the bona fide mating agglutination, thus we name this behavior as pseudomating. Hence, the pseudomating is a possible assay to evaluate the bioactivity of the ciliary ectosomes. Thus, another interesting question is whether the activities of the ectosomes from the TZ mutants were altered. To avoid possible

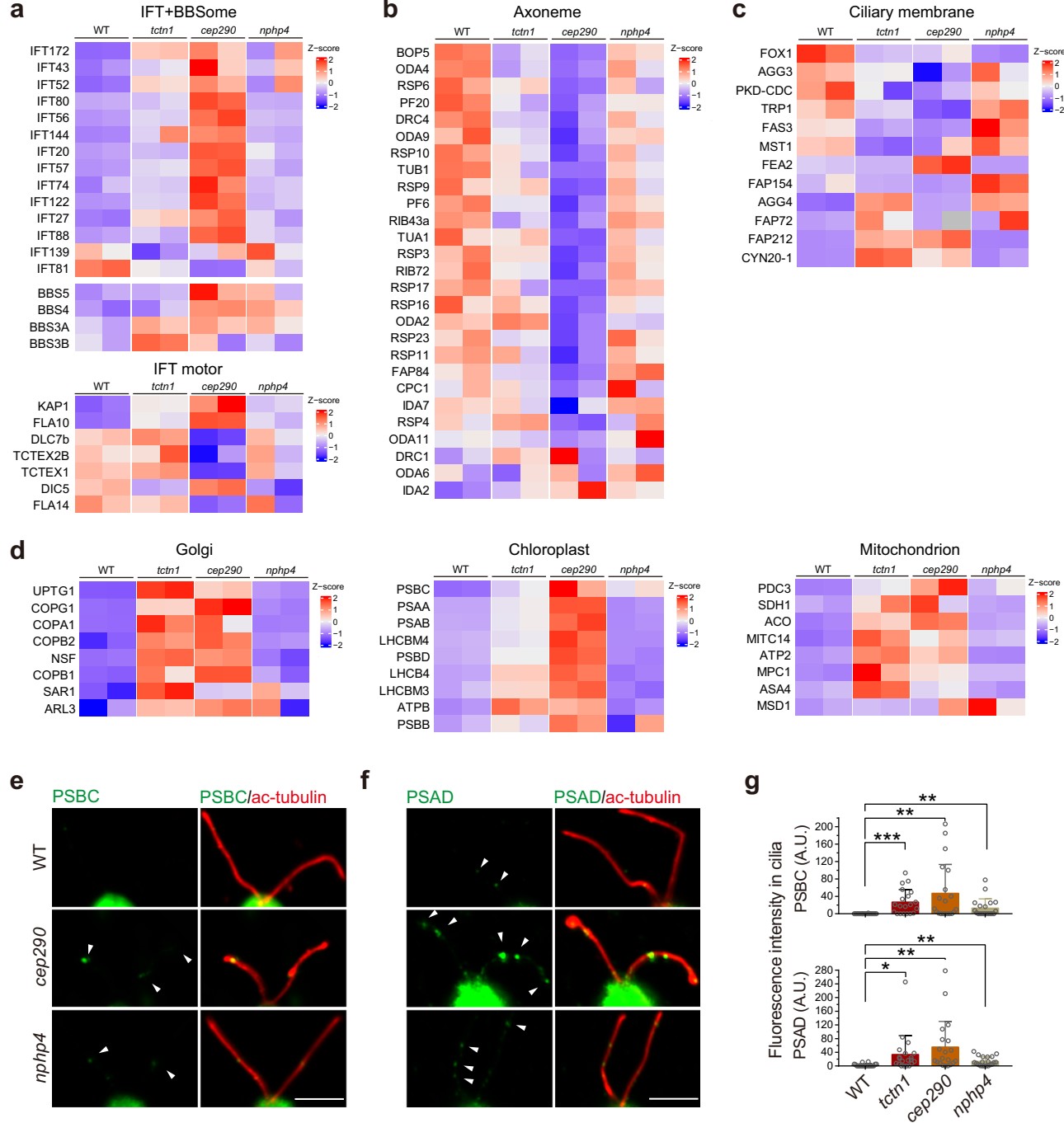

**Fig. 5 Proteins differentially regulated in cilia by different TZ proteins. a–d** Heatmap of representative groups of proteins identified from ciliary proteomics analysis of cilia from WT, *tctn1*, *cep290*, and *nphp4* cells (two replicates for each strain, column). Representative proteins were selected from the mass spectrometry data in Supplementary Fig. 6a and grouped into IFT and BBSome (**a**, top), IFT motor (**a**, bottom), Axoneme (**b**), Ciliary membrane (**c**), Golgi (**d**, left), Chloroplast (**d**, middle), and Mitochondrion (**d**, right) proteins. The *Z*–score is shown by color key in the heatmaps. **e, f** Immunostaining images displaying the enrichment of photosystem proteins (PSBC and PSAD) in TZ mutants (*cep290* and *nphp4*). WT and TZ mutants were immunostained with anti-PSBC (**e**, green), or anti-PSAD (**f**, green) and anti-acetylated α-tubulin (ac-tubulin, red) antibodies. The non-ciliary proteins were transported and accumulated in cilia of TZ mutant cells. The arrowheads indicate the accumulations of photosystem proteins in the cilium. Scale bar, 5 µm. **g** Bar–scatter graphs show the ciliary fluorescence intensity (arbitrary units, A.U.) of PSBC or PSAD. Data are the mean ± SD (*n* = 20). Statistical significance was determined with an unpaired *t* test. \**P* < 0.05; \*\**P* < 0.01; \*\*\**P* < 0.001 by two tailed. Source data are provided as a Source Data file.

contamination with cell debris that may be pelleted during ultracentrifugation, we performed ectosome purification using a commercial exosome/ectosome isolation kit, which specifically precipitated the membrane vesicles upon relatively low-speed centrifugation (at 10,000×*g*). As denoted in the "Methods"

section, we conducted a gamete pseudomating assay with the fractions of S20, Sup, and Eco, and found that the purified ectosomes, but not the supernatant fraction, possessed the ability to induce the adherence of 6145c gametic cells, not 6145c vegetative cells or 6145c gametic cells alone (Fig. 7a, b). After mixing

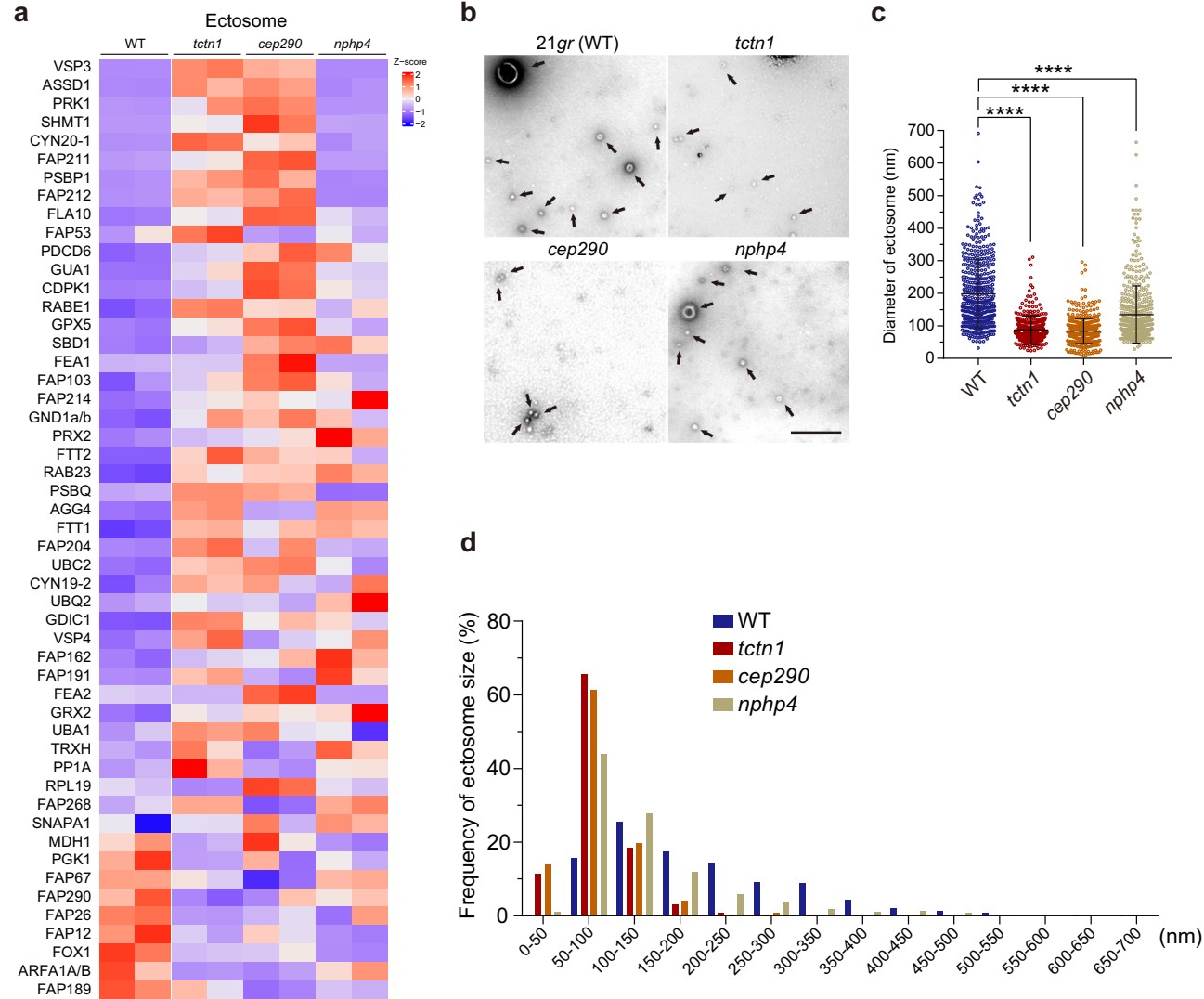

**Fig. 6 The formation of ciliary ectosomes is regulated by the integrity of the TZ during gamete mating events. a** Heatmap of representative ectosome-related proteins identified from ciliary proteomics analysis of cilia from WT, *tctn1*, *cep290*, and *nphp4* cells (two replicates for each strain, column). Representative proteins were selected from the mass spectrometry data in Supplementary Fig. 6a. The *Z*–score is shown by color key in the heatmap. **b** Negative stain transmission electron micrographs of purified ciliary ectosomes (black arrows) from gametes in the mating process. 21*gr* (WT), ectosomes from mating supernatant of 21*gr* × 6145c; *tctn1*, ectosomes from mating supernatant of *tctn1* × 6145c; *cep290*, ectosomes from mating supernatant of *cep290* × 6145c; *nphp4*, ectosomes from mating supernatant of *nphp4* × 6145c. Scale bar, 500 nm. **c** Scatter plot showing the diameters of ectosomes shedded from the cilia of different mating gametes. The sizes of the ectosomes are the mean ± SD ($n = 500$) described in **b**. Statistical significance was determined with an unpaired *t* test. ****$P < 0.0001$ by two tailed. **d** Graph showing the distribution of ciliary ectosomes of different sizes purified from 21*gr* × 6145c, *tctn1* × 6145c, *cep290* × 6145c, and *nphp4* × 6145 described in **b**. Source data are provided as a Source Data file.

with the ectosomes for 10 min, we measured the percentages of the adhering minus gametes (6145c) and found that the efficiency of the ectosomes from the TZ mutants, especially from the *tctn1* or *cep290* group, decreased dramatically (Fig. 7c). Consistent with Fig. 7b, the ectosomes failed to induce agglutination of vegetative 6145c cells in Con group (Fig. 7c). As expected, SDS–PAGE with silver staining showed variations in the protein amount and composition of ectosomes from the four groups (Fig. 7d and Supplementary Fig. 7a). Thus, these data further demonstrate that the TZ also plays roles in regulation of the biological activity and protein composition of ciliary ectosomes.

## Discussion

As the only transport route from the cell body to the cilium, the TZ compartment plays essential roles in the regulation of ciliary composition, which fundamentally defines the biological

processes and functions of the cilium. Although the structure and composition of the TZ have been characterized, the unique roles of TZ proteins in assembly of the TZ and their systematic regulation of ciliary components are largely unknown. Consistent with previous studies in mammals and worms[9,31,36,45–50], we found that *C. reinhardtii* TCTN1 localizes in the TZ (Fig. 1i, j). Furthermore, we provide both ultrastructural insights and systematic proteomic information upon the loss of TCTN1. Using immunofluorescence microscopy and Immunogold TEM, we showed that TCTN1 localized in the middle region of the TZ independent of CEP290 and NPHP4 (Fig. 3a–e). Using TEM, we found that TCTN1 is essential for the assembly of wedge-shaped structures, with a phenotype similar to that of the *nphp4* mutant (Fig. 2b, c)[40]. Previous studies indicated that TCTN1 is located at the ciliary membrane and CEP290 is located near the axoneme[51]. Based on our data, we speculate that TCTN1 and CEP290 may

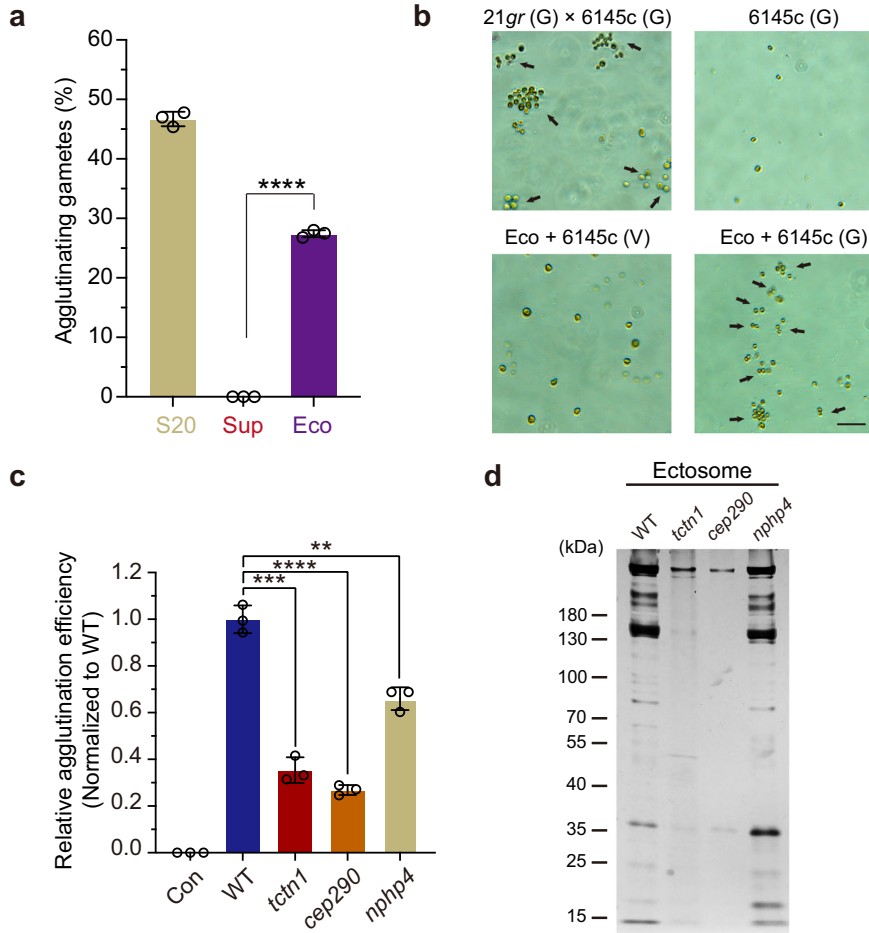

**Fig. 7 The activity and protein composition of ciliary ectosomes shedding from gametes of different TZ mutants are different. a** Bar–scatter graph depicting the agglutinating gametes number (%) from 6145c (mt-) mixed with the supernatant (S20), the final supernatant (Sup), and ectosome pellet (Eco) from gamete mating culture (21*gr* × 6145c). **b** PH images showing the agglutination of gametic 6145c (mt-) induced by mixing with the ectosomes (Eco). Gametic 6145c alone, vegetative 6145c with Eco, and 21*gr* × 6145c were showed as the negative and positive controls, respectively. The black arrows indicate the agglutination of gametic cells. G gametic cells, V vegetative cells. Scale bar, 20 μm. **c** Bar–scatter graph displaying the agglutination efficiency (normalized to WT) of gametic 6145c (mt-) mixed with the ectosome pellet (Eco) from gamete mating cultures (21*gr* × 6145c, *tctn1* × 6145c, *cep290* × 6145c, *nphp4* × 6145c). The group of vegetative 6145c with Eco was as the negative control (Con). **d** Silver-stained SDS–PAGE gel showing protein variations in ectosomes isolated from the plus mating type gamete (21*gr*, *tctn1*, *cep290*, *nphp4*) with minus mating type gamete (6145c). The indicated ciliary ectosomes purified from the same amount of cells were loaded with equal number of mating gametes, separated on 4–20% SDS–PAGE and then visualized by silver staining. The positions of standard proteins and their molecular masses in kDa are indicated. Data are presented as the mean ± SD (*n* = 3, independent experiments) in this figure. Statistical significance was determined with an unpaired *t* test in this figure. **P < 0.01; ***P < 0.001; ****P < 0.0001 by two tailed. Source data are provided as a Source Data file.

function as anchors of connection structure at the two ends of the TZ, and loss of either of the two key components disrupts the structure. In support of this speculation, we found that the distance between the ciliary membrane and axoneme is longer in *tctn1* cells than in WT cells. Another interesting observation in this paper is that loss of TCTN1 leads to ciliary glycocalyx malformation, which is also seen in the *FMG-1b* mutant[52]. Whether loss of ciliary glycocalyx plays a role in ciliary ectosome shedding will be an interesting question for future study.

The TZ strictly regulates the entry and retention of ciliary proteins and exclusion of nonciliary proteins[9,34,47,53,54]. It is known that the severity of TZ defects caused by mutations in individual TZ proteins varies, reflecting the different functions of individual proteins in organizing the TZ. Consistently, our phenotypic analysis of the three TZ mutants (*tctn1*, *cep290*, and *nphp4*) also showed various ciliary phenotypes (Supplementary Fig. 3a–d), which were caused by the different systematic changes in the ciliary components (Supplementary Fig. 6a). To date, how

the loss of a single TZ protein systematically affects ciliary composition is not clear. Taking advantage of cilia purification from *Chlamydomonas*, we isolated the cilia of WT, *tctn1*, *cep290*, and *nphp4* cells and subjected the ciliary samples to proteomic analysis. In agreement with the variable phenotypes, the proteomic profile of cilia from each cell type was quite unique (Fig. 5a–d and Supplementary Fig. 6a). The concentrations of ciliary membrane-associated proteins, including PKD, FOX1, AGG4, and CYN20-1, were disrupted in *tctn1*, *cep290*, and *nphp4* mutant cilia. Considering the function of membrane proteins, ciliary signaling should be misregulated. Although mutants displayed characteristic protein profiles and the protein ratios were variable in different mutants, we indeed observed several common features shared by the three TZ mutants. First, the proteomic data showed accumulation of IFT subunits and several IFT motors and reduction of the axonemal components in the mutant cilia. Second, nonciliary proteins, including various ATP synthases (ATP1A, ATP2, ATPC, and ASA4), photosystem

proteins (PSAD, PABC, PSAA, and PSAB), vesicular transport proteins (Coatomer subunits), and translation initiation factors (EIF3 subunits), also accumulated to different degrees, which is direct evidence of the function of the TZ to exclude nonciliary components. However, thus far, we do not know how these nonciliary proteins could pass through the ciliary gate and whether they have functions in cilia. Furthermore, according to the above phenotypic analysis and ciliary protein profiling in these three TZ mutants (*tctn1*, *cep290*, *nphp4*), we conclude that the different roles of TCTN1, CEP290 and NPHP4 in protein sorting ultimately influence the phenotype. The hierarchy of degrees of phenotypic severity in the three TZ genes is *CEP290 > TCTN1 > NPHP4*. How they coordinately influence ciliary composition and ciliary functions remains to be further investigated.

Another interesting phenomenon observed in our data inspire us that there may be a link between the TZ proteins and ciliary ectosomes. First, the cellular morphology of TZ mutants (*tctn1* and *cep290*) are palmelloid (Fig. 1c and Supplementary Fig. 3a)[34], indicating the deficient of ciliary ectosome associated lysis protease shedding within the mother cell walls after cell division[19,26]. Second, the proteomic analysis was changes in ciliary ectosome-associated proteins (e.g., FOX1, AGG4, CYN20-1) (Fig. 6a)[20]. Until now, it has been unknown whether the TZ structure controls the formation of ciliary ectosomes. Here, we provide direct evidence that defines the function of the TZ in ectosome biogenesis. Loss of the integrity of the TZ affects the morphology, protein composition, and biological activities of ciliary ectosomes (Figs. 6 and 7). In the future, it will be worth investigating the mechanisms through which TZ proteins influence the amount of ectosome-associated proteins in cilia and the control of ectosome formation and biological activity.

In summary, our work showed that TCTN1 is essential for proper assembly of the ultrastructure of the TZ. Using proteomic analysis of cilia from WT, *tctn1*, *cep290*, and *nphp4* cells, we systematically characterized the ciliary protein profile of the mutants, which revealed the unique function of each TZ protein. Most importantly, a function of the TZ in ciliary ectosome formation was uncovered.

## Methods

**Strains and cell culture.** *C. reinhardtii* wild-type strains 21*gr* (CC-1690, wild-type, mt+) and 6145c (CC-2895, wild-type, mt−) and the mutant strains *cep290* (CC-4374, mt+) and *nphp4* (CC-5113, mt+) were provided by the *Chlamydomonas* Resource Center (University of Minnesota, USA). The *tctn1* mutant was generated from the wild-type strain 21*gr* by insertional mutagenesis with the paromomycin-resistant DNA fragment *aphVIII*. Strains were cultured in Tris–acetate–phosphate (TAP) plates or liquid medium with aeration at 23 ± 0.5 °C with a light/dark cycle of 14/10 h at a light intensity of 8000 lx. For transformation experiments, the indicated cell line was grown in TAP liquid medium under continuous light. Autolysin was used to release *tctn1* and *cep290* mutant cells from the mother cell wall and to induce ciliogenesis. For the induction of gametogenesis, vegetatively growing cells were transferred to N-free medium as previously described[22]. For *tctn1* or *cep290* gametes, autolysin was applied to induce the release of gametes from the palmelloid before mating. The cells were thoroughly washed to remove autolysin and allowed to assemble cilia for ~2 h.

**DNA manipulations, transformation, and mutagenesis.** A library of mutants was generated via random insertional using electroporation with the *aphVIII* gene (paromomycin-resistant cassette, ~2.1 kb fragment cut with *EcoRI* from the plasmid pJMG-aphVIII)[55]. Electroporation was performed as described previously[55]. Briefly, cells for transformation were cultured in liquid TAP medium with constant aeration and continuous light until the cell concentration reached ~1.0 × 10⁷ cells/mL. Then, the cells were inoculated into fresh liquid TAP medium and grown under continuous light for 18–20 h until the cell concentration reached ~4.0 × 10⁶ cells/mL. Cells were transformed with the DNA fragment via square-wave electroporation with a BTX ECM830 electroporation apparatus (500 V, 4 ms, 6 pulses) (BTX, USA). The flanking sequences of the *aphVIII* insertional site were determined by restriction enzyme site-directed PCR (RESDA-PCR)[56], sequencing and blasting with *Chlamydomonas reinhardtii* v5.6 genome database (https://phytozome-next.jgi.doe.gov/info/Creinhardtii_v5_6). For the gene complementation experiments, full-length DNA harboring the *TCTN1* locus (with an endogenous promoter) was

inserted into the expression construct pHyg-3HA. The resulting construct was then linearized with *NdeI* and transformed into *tctn1* mutant with electroporation. The transformants were plated onto the hygromycin (Cat. no. V900372, Sigma, USA) plates and screened for the ciliary phenotypes and via immunoblot analysis.

**Analysis of ciliary and cellular phenotypes.** The liquid-medium-cultured cells were put on the slide, and observed the cell status with inverted microscope. To capture the cell images, cells were fixed with 5% glutaraldehyde solution, plated on the slide with a cover-slip, observed and captured by LAS V4.0 imaging software on a Leica DMI4000 B Inverted Microscope (Leica, Germany) with a ×10, ×20, ×40, or ×63 objective (HCX PL FLUOTAR ×10/0.30, 506507; HCX PL FLUOTAR L ×20/0.40, 506243; HCX PL FLUOTAR L ×40/0.60, 506203; HCX PL FLUOTAR L ×63/0.70, 506217) for differential interference contrast (DIC)/phase contrast (PH) images. The ciliary length was measured for 50 cells by ImageJ (National Institutes of Health, USA). The cell number was counted for 200 cells. All original data were analyzed and plotted with GraphPad Prism (GraphPad Software, USA). All experiments were repeated three times.

**Ciliary deciliation, regeneration, and resorption.** To induce the ciliary deciliation, cells was deciliated by pH shock method[57,58]. Briefly, cells were quickly treated with 0.5 M glacial acetic acid, kept at pH ~4.5 for 30 s, then quickly added by 0.5 M KOH to recover the pH ~7.0. To induce regeneration of the cilia, cells after deciliation were centrifuged and washed with fresh M medium to be allowed for ciliary regeneration with aeration at 23 ± 0.5 °C in light. To induce resorption of the cilia, cells incubated with 20 mM sodium pyrophosphate (NaPPi) in M medium gradually shorten cilia over time[59]. At the indicated time point, cells were collected for imaging and measurement of ciliary length or for the immunoblot analysis. All experiments were repeated three times.

**Autolysin preparation.** The cell wall proteolytic enzyme autolysin is typically generated during plus and minus gametes mating[27]. Generally, we followed the gamete autolysin preparation procedure (https://www.chlamycollection.org/methods/preparation-of-gamete-autolysin/), except M-N medium for nitrogen starvation was used to induce gamete differentiation. Then cells at the density of ~3.0 × 10⁶ cells/mL were incubated in M-N medium for 20–24 h under continuous light (8000 lx). Equal amount of opposite mating type gametes (21*gr* × 6145c) were mixed to allow mating with gentle aeration for ~15 min. After centrifugation at 600×*g*, for 5 min and 14,000×*g*, for 10 min, respectively, the final supernatant was filtrated by 0.22 μm membrane filter and aliquot, stored at −80 °C until use.

**Isolation and fractionation of cilia and ciliary ectosomes.** To isolate cilia, cells were deciliated by pH shock. Detached cilia were enriched via sucrose gradient centrifugation and pelleted by high-speed centrifugation[60]. To fractionate the ciliary samples into the M + M (membrane and matrix) fraction and Axo (axoneme) fraction, the ciliary pellet was lysed in 0.5% NP40 buffer A (50 mM Tris–HCl [pH 7.5], 10 mM MgCl₂, 1 mM EDTA, 1 mM DTT) with EDTA-free protease inhibitor cocktail (Cat. no. 11836170001, Roche, Switzerland), frozen in liquid nitrogen and thawed, followed by centrifugation at 20,000×*g* at 4 °C for 10 min. The supernatants and pellets were the M + M fraction and Axo fraction, respectively.

Isolation of ciliary ectosomes by ultracentrifugation was carried out as described previously[22] with slight modifications. Briefly, The mating type plus (mt+) gametes (21*gr*, *tctn1*, *cep290*, and *nphp4*) are mixed with the mating type minus (mt−) gametes 6145c (mt+: mt− = 5: 1) for 15 min, followed by 600×*g* for 5 min, 20,000×*g* for 10 min, and the final step of 150,000×*g*, 60 min to obtain the ectosome pellet (Eco). The ciliary ectosomes were also purified with a Total Exosome Isolation (from cell culture media) Kit (Cat. no. 4478359, Thermo Fisher, USA). The media with above described mating cells were centrifuged at 600×*g* for 5 min and 20,000×*g* for 5 min (3 times) to remove the cells and debris. The resulting supernatant (S20) was finally incubated in buffer at 4 °C for 2 h and then pelleted by centrifugation at 10,000×*g* for 60 min to obtain the final supernatant (Sup) and ectosome pellet (Eco). The pelleted ectosomes were resuspended in N-free medium. The ectosomes were image by TEM. At least 500 ectosomes were measured for their diameters by ImageJ software for each group. The experiments were performed for three times.

**Assay for ectosome activity.** To determine the activity of ectosomes, purified ectosomes were applied for gamete agglutination with 6145c. In particular, ectosomes from different mating events (21*gr*, *tctn1*, *cep290*, or *nphp4* × 6145c) were mixed with 6145c gametes (10 μL, 1.0 × 10⁸ cells/mL) to induce agglutination for 10 min. The agglutinating gametes were observed and imaged alive. The ratios of cell adhesion were plotted using GraphPad Prism software (GraphPad Software, USA).

**Immunofluorescence microscopy.** Immunofluorescence was carried out essentially as previously described[57]. For detection of CEP290 and NPHP4, cells were collected and resuspended in MT buffer (30 mM HEPES [pH 7.2], 3 mM EGTA, 1 mM MgSO₄, 25 mM KCl), followed by fixation and extraction in 100% prechilled

methanol for 20 min at −20 °C. For detection of other proteins, cells were collected and resuspended in MT buffer for 2 min, followed by fixation in 4% paraformaldehyde fixation solution (Cat. no. BL539A, Biosharp, China) for 5 min at room temperature. After the fixation step, cells were resuspended in MT buffer with 0.5% NP-40 for 2 min and further extracted in 100% prechilled methanol for 10–15 min at −20 °C. The cells were then sequentially rehydrated with PBS, blocked with goat blocking buffer (containing 5% goat serum) (Cat. no. E674004, Cat. no. E510009, Sangon Biotech, China), and incubated with primary antibodies at 4 °C overnight, followed by incubation with secondary antibodies at 37 °C for 2 h.

The secondary antibodies used were preadsorbed anti-rat IgG H&L (Alexa Fluor® 488) (Cat. no. ab150165), preadsorbed anti-mouse IgG H&L (Alexa Fluor® 594) (Cat. no. ab150120), preadsorbed anti-rabbit IgG H&L (Alexa Fluor® 647) (Cat. no. ab150087), preadsorbed anti-rabbit IgG H&L (Alexa Fluor® 594) (Cat. no. ab150084), preadsorbed anti-rabbit IgG H&L (Alexa Fluor® 488) (Cat. no. ab150077), and preadsorbed anti-mouse IgG H&L (Alexa Fluor® 647) (Cat. no. ab150119) (Abcam, UK). The secondary antibodies were used at a dilution of 1:500.

After three washes for 5 min with 0.5% Tween-20 PBS, one wash with PBS, and one wash with Milli-Q, samples on the slides were mounted with DAPI-Fluoromount-G (Cat. no. 0100-20, SouthernBiotech, USA), sealed with nail polish and air dried for ~2 h before observation. Samples were then imaged on a Leica TCS SP5 II or Leica TCS SP8 confocal laser microscope (Leica, Germany) with a ×63 oil immersion objective (HCX PL APO ×63/1.40–0.60 OIL CS, 506188). Structured Illumination Microscopy (SIM) super-resolution images were acquired using an N-SIM Super-Resolution Microscope System (Nikon, Japan) equipped with a ×100 oil immersion objective (SR HP Apo TIRF 100× H/1.49). Raw image data were captured and processed in Leica Application Suite X (Leica, Germany) or NIS-Elements AR (Nikon, Japan), Adobe Photoshop CC (Adobe Systems Incorporated, USA), and assembled using Adobe Illustrator CC (Adobe Systems Incorporated, USA).

**SDS–PAGE and immunoblotting.** Cells were harvested by centrifugation at 10,000×g for 1 min, frozen in liquid nitrogen and stored at −80 °C until use. The frozen samples were then lysed in prechilled buffer A (50 mM Tris–HCl [pH 7.5], 10 mM MgCl₂, 1 mM EDTA, 1 mM DTT) with EDTA-free protease inhibitor cocktail (Cat. no. 11836170001, Roche, Switzerland) and boiled in 1× SDS sample buffer for 5 min before being subjected to SDS–PAGE and blotting analysis. The secondary antibodies used were HRP-conjugated goat anti-mouse, goat anti-rabbit and goat anti-rat (1:5000; Cat. no. 115-035-003, Cat. no. 111-035-003, Cat. no. 112-035-003, Jackson, USA). The final immunoreactive bands were visualized and captured using an Amersham imager 600 Control (GE, USA).

**Primary antibodies.** The primary antibodies used for immunoblotting (IB) or immunofluorescence (IF) were as follows: anti-HA high affinity (clone 3F10, rat monoclonal IgG, 1:3000 for IB and 1:50 for IF; Cat. no. 11867423001, Roche, Switzerland), anti-α-tubulin (clone 1E4C11, mouse monoclonal IgG, 1:5000 for IB and 1:200 for IF; Cat. no. 66031-1-Ig, Proteintech, USA), anti-α-tubulin (rabbit polyclonal IgG, 1:5000 for IB; Cat. no. 11224-1-AP, Proteintech, USA), anti-centrin (clone 20H5, mouse monoclonal IgG, 1:400 for IF; Cat. no. 04-1624, Merck Millipore, Germany), anti-acetylated α-tubulin (clone 6-11B-1, mouse monoclonal IgG, 1:200 for IF; Cat. no. T7451, Sigma, USA), anti-CEP290 (rabbit polyclonal IgG, 1:200 for IF; a gift from Dr. George Witman), anti-NPHP4 (rabbit polyclonal IgG, 1:100 for IF; a gift from Dr. George Witman), anti-FMG-1B (clone SP2/0, mouse monoclonal IgG, 1:100 for IB; Cat. no. AB_2722112, Developmental Studies Hybridoma Bank), anti-IFT122 (rabbit polyclonal IgG, 1:2000 for IB and 1:100 for IF)[61], anti-IFT121 (rabbit polyclonal serum, 1:2000 for IB)[62], anti-IFT172 (rabbit polyclonal IgG, 1:2000 for IB)[62], anti-IFT57 (rabbit polyclonal serum, 1:2000 for IB and 1:100 for IF)[63], anti-IFT54 (rabbit polyclonal serum, 1:2000 for IB)[61], anti-IFT43 (rabbit polyclonal serum, 1:100 for IF)[61], anti-IFT38 (rabbit polyclonal IgG, 1:5000 for IB and 1:100 for IF)[62], anti-FLA10 (rabbit polyclonal IgG, 1:3000 for IB)[62], anti-D1BLIC (rabbit polyclonal IgG, 1:2000 for IB)[62], anti-IC2/IC69 (clone 1869A, mouse monoclonal IgG, 1:20,000 for IB; Cat. No. D6168, Sigma, USA)[62], anti-PSAD (rabbit polyclonal serum, 1:1000 for IB, 1:500 for IF; a gift from Dr. Xiaobo Li; Cat. no. AS09461, Agrisera, Sweden), and anti-PSBC (rabbit polyclonal serum, 1:3000 for IB, 1:500 for IF; a gift from Dr. Xiaobo Li; Cat. no. AS111787, Agrisera, Sweden). Rabbit anti-RIB72 antibodies were generated against bacterial expressed GST-His-tagged RIB72 (25–290 amino acids) (Abclonal, China) and were only used at a dilution of 1:3000 for IB. Rabbit anti-BBS8 antibodies were generated against bacterial expressed GST-His-tagged BBS8 (16–193 amino acids) (Abclonal, China) and were only used at a dilution of 1:250 for IB. Rabbit anti-TCTN1 antibodies were generated against bacterial expressed GST-His-tagged TCTN1 (368–490 amino acids) (Abclonal, China) and were only used at a dilution of 1:500 for IF.

**Transmission electron microscopy (TEM).** Negative staining and TEM were carried out as previously described[22,60]. For immuno-EM, cells were fixed by high-pressure freezer (Leica, HPM100). The sections were washed by 100 mM phosphate buffer (PB, pH = 7.2) for 1 min and then blocked in 2% BSA, 0.1% tween-20, and 0.1% Cold Fish Gelatin in PB for 30 min. After blocking, the samples were labeled with the primary antibody (1:10, rabbit monoclonal anti-HA, clone C29F4, Cat. no. 3724, CST, USA) at 4 °C overnight. After washing 5 times by PBG (0.1% Gelatin in PB), the samples were incubated with gold-conjugated secondary antibody (1:10, anti-rabbit gold conjugant, 10 nm, Cat. no. G7402, Sigma, USA) for 2 h at room temperature. After washing 5 times by PBG and 2 times by PB, the samples were fixed in 1% glutaraldehyde for 5 min and washed 3 times by distilled water. Sections were stained with uranyl acetate. The samples were imaged on an H-7650B transmission electron microscope (Hitachi Limited, Japan) equipped with a digital V600 camera (ATM Company).

**Quantitative proteomics.** A total of 150 μg of ciliary proteins from each sample was dissolved in RIPA buffer containing protease inhibitor. The dissolved proteins were precipitated and washed with acetone solution. The concentration of redissolved samples was determined with a bicinchoninic acid protein quantification kit (Cat. no. 23225, Fisher Scientific, MA, USA). Then, 100 μg of protein was reduced with 5 mM dithiothreitol and alkylated with 10 mM iodoacetamide in 100 mM HEPES with 1% SDC. Trypsin (1 μg) was used for protein digestion overnight at 37 °C. SDC and salt were removed, and the peptides were dried in a freeze dryer. The peptides, resuspended in 0.1% TFA, were separated using a Reprosil-Pur 120 C18 analytical column (100 μm ID × 15 cm, 1.9 μm, Dr. Maisch) on a nano-UPLC system (EASY-nLC1200). Then, 0.1% formic acid in acetonitrile/water (2:98) was used as mobile phase A, and 0.1% formic acid in acetonitrile/water (80:20) was used as mobile phase B. A Q-Exactive HFX mass spectrometer (Thermo Scientific, MA, USA) was operated in data-dependent acquisition mode for sample analysis. MS1 was performed in a range of 350–1600 $m/z$ with a resolution of 120,000 (200 $m/z$). The top 20 precursor ions were fragmented by high-energy C-trap dissociation (HCD) with a normalized collision energy (NCE) of 27%. In MS/MS, the resolution was set to 15,000, the AGC control was $1.0 \times 10^5$, the maximum ion introduction time was 110 ms, and the dynamic exclusion time was 45 s. The MS/MS spectra from each run were searched against the species-level UniProt FASTA databases (*Chlamydomonas reinhardtii*, 2020-01-11, total entries 31247, reviewed entries 339, unreviewed entries 30908). The raw data were processed using Proteome Discoverer (PD) software. The search criteria were as follows: tryptic digestion, 2 missed cleavages allowed, carbamidomethyl (C) set as a fixed modification, and oxidation (M) and acetyl (protein N-term) set as variable modifications. Peptide identification was carried out with an initial precursor ion mass deviation of up to 10 ppm and a fragment mass deviation of 0.02 Da. For protein identification, the false discovery rate (FDR) was set at 0.01 for both peptide spectral matches (PSMs) and peptide levels. Other parameters were set as default. Label-free quantification was performed by the summed abundances of the top 3 peptides. Both of unique peptides and razor peptides were used for label-free quantification. Peptide abundance was based on intensity of corresponding precursor ion. Quantitative protein data were further normalized by total peptide amount. Raw mass spectrometry data have been deposited to the ProteomeXchange Consortium via the Proteomics Identification Database (PRIDE) partner repository[64].

Volcano plot and heatmap plots were used to visualize the quantitative data. The volcano plot was constructed using $-\log_{10}$ ($P$ value) against $\log_2$ (fold change value of *tctn1*/WT). The red and blue dots represent the upregulated and downregulated proteins in cilia of the *tctn1* mutant. The nonaxial vertical lines indicate fold change of *tctn1*/WT < 0.5 and >2. The nonaxial horizontal line represents a $P$ value of 0.05. To construct the heatmap plots, the abundance value of each sample was listed in a matrix, in which each row represented a protein and each column represented a set of samples. The average and standard deviation were calculated for each protein. The spread was obtained using the protein abundance value of each sample minus the average. Z-score was obtained by dividing each spread by the standard deviation of each protein. Next, the normalized Z-scores were depicted by blue–white–red color schemes to visualize the protein expression levels in the cilia of WT and mutant cells.

**Statistics and reproducibility.** All statistical tests were performed in GraphPad Prism. The number of independent experiments or samples is listed in the figure legends. All data are expressed as mean ± SD or mean ± SEM values as indicated in the figure legends. Two-way ANOVA with Dunnett's test or unpaired t test with two-tailed were applied for comparing differences among multiple groups or between two groups, respectively. Significance was considered if the $P$ value was <0.05. No data were excluded from the analyses. Microscopic and biochemical assays were independently repeated at least three times.

**Reporting summary.** Further information on research design is available in the Nature Research Reporting Summary linked to this article.

## Data availability

The mass spectrometry proteomics data have been deposited to the ProteomeXchange Consortium via the PRIDE partner repository with the dataset identifier PXD033090. The accessible hyperlink of *Chlamydomonas reinhardtii* v5.6 genome database is https://phytozome-next.jgi.doe.gov/info/Creinhardtii_v5_6. All other raw data and the biological

materials that support the findings of this study are available from the corresponding author upon reasonable request. Source data are provided with this paper.

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

## Acknowledgements

We thank Dr. George Witman and Dr. Xiaobo Li for kindly providing antibodies. We would also like to thank Dr. Ying Li, Dr. Jingli Hou, Dr. Rongpeng Li, Dr. Jun Lu, and Dr. Yuanlin Zheng for technical and equipment support. We are grateful to Cell biology facility and the Center of Biomedical Analysis (Tsinghua University, China) and Instrumental Analysis Center (Shanghai Jiao Tong University, China) for their excellent help. This work was supported by the National Key Research and Development Program of China (2021YFC2700800), the National Natural Science Foundation of China (91954123, 31972887, 31991191, and 31972888), the Natural Science Foundation of Jiangsu Province (BK20191465), Clinical Research Projects of Shanghai Municipal Health Commission (20194Y0133), Shanghai Science and Technology Commission (20JC1410100), Innovative research team of high-level local universities in Shanghai (SHSMU-ZDCX20211800), and the Priority Academic Program Development of Jiangsu Higher Education Institutions (PAPD).

## Author contributions

L.W., X.W., and M.C. designed, performed the experiments, and analyzed the data. Z.W., Z.L., C.L., H.Z., H.Y., Y.L., and Y. Cheng performed the experiments. Y. C. and G.L. analyzed and interpreted the data. J.P., L.W., and M.C. conceived the idea, analyzed the data and wrote the manuscript. All authors approved the final manuscript.

## Competing interests

The authors declare no competing interests.
