## [Peer Review File · Nature Communications]

Ciliary transition zone proteins coordinate ciliary protein composition and ectosome sheddingREVIEWER COMMENTS

Reviewer #1 (Remarks to the Author):

Authors Wang et al propose in their manuscript "Ciliary transition zone proteins coordinate ciliary protein composition and ectosome shedding" a role for the transition zone (TZ) protein TCTN1. In this review, I comment primarily on the mass spectrometry proteomics used. This work in part uses mass spectrometry proteomics to identify candidate proteins that are differentially abundant in WT vs *tctn1* cilia. They then validate the mass spectrometry findings using immunofluorescence and immunoblotting experiments. Then, further mass spectrometry experiments are used for cilia purified from WT, *tctn1*, *cep290*, and *nphp4* cells. These experiments were used to compare the differential protein abundances across the groups.

The authors have written a concise, easy to read research article. I found the rationale of their experiments clearly communicated and easy to follow, and I also appreciate that the interpretation of their results was concise and to the point. Overall, it made the manuscript easy to read and I look forward to its publication in Nature Communications as a broadly interesting investigation of the TZ protein TCTN1 and its role in ciliary formation and ectosome shedding.

I have just a couple minor suggestions to improve the repeatability and clarity of the work, which are detailed below.

- Please deposit the RAW mass spectrometry data in a repository, e.g. MASSIVE, PRIDE, etc.
- Figures 4, 5, 6, and S6 all use "fold change" as a measure of abundance, however there is no detail how this was calculated in the Methods. Please detail how the fold change was calculated (e.g. any data normalizations or imputations) in the Methods section and label these axes and/or captions with the fold change details (e.g. log2?) so that the figures are more clear.

Reviewer #2 (Remarks to the Author):

Here Wang and colleagues performed genetic screening to identify genes related to ciliogenesis using *Chlamydomonas reinhardtii* as a model system and reported the role of *tctn1* at the ciliary transition zone, together with previously characterized *cep290* and *nphp4* at the same location, for transition gating to control ciliary proteome composition. Although *tctn1* was reported for its association to human ciliopathy Meckel and Joubert syndrome over ten years ago (Garcia-Gonzalo, et al., 2011), we still do not fully understand how the proteins at the transition zone regulate ciliary compositions. This manuscript claimed that *tctn1* has distinctive functions in transition gating compared to two known complexes (MKS complex and NPHP complex) and is also involved in determining the ciliary ectosome composition. In my opinion, proteomics analysis is critical for authors to claim the distinctive role of *tctn1* compared to *cep290* and *nphp4*, mainly showing the different ciliary proteome compositions in each condition. However, the authors did not present all information correctly, so I cannot follow this conclusion based on the results presented here.

[Major issues]

(1) All heatmap plots were not clear what they were presented. The authors described it as "relative fold-changes" but did not mention "the reference." Is it based on the protein abundance of wild-type samples (I doubt it, because wild-type samples also have varied signals)? Or the mean of all samples (how can I interpret this "fold-change")? Without proper reference, I cannot interpret any of these data.

(2) In SFig6 (I think it is "the comprehensive raw data for the main figures"), almost all proteins seem to be quite different in all conditions. Not sure about *cep290* and *nphp4* mutants, but at least *tctn1* mutants do not have that severe defect on cilia formation (after complete deciliation, the cilia is

generally recovered only with shorter length). So I expected most ciliary proteomes to be preserved even in the *tctn1* mutant, but this figure did not look like that. The authors need to clarify this (unless authors claimed that all > 2,000 observed proteins were significantly changed across three mutants).

(3) Fig 4c showed the change of fold change in cilia between WT and *tct1* mutant, but again it is not clear how the authors calculated them. Like Fig 4b and Fig 4e, if the authors compared it to the wild-type cilia proteome, green should be zero, but it does not look like that. Also, based on Fig 5 and Fig 6, I guess that the authors have duplicate data. If true, these values should be presented individually rather than the mean and the error bars.

(4) Because the authors also claimed the ectosome shedding, it would be critical to control the background/contamination of ciliary proteome. I think it is necessary to analyze the proteome in the media or similar samples to rule out the proteins not available in the ciliary structure.

[Minor issues]

(1) There is no description of how the authors quantify the protein abundances. The authors only described "Total peptide was used for normalization, and both unique peptides and razor peptides were used for further quantification" in the method. Is it based on spectral counting or ion intensity? Did the authors use the internal spike-in? More details about proteomic analysis should be provided.

(2) The RAW data for proteomics were not deposited to the repository, like the PRIDE database. I believe that it is also essential for the reproducibility and transparency that this journal pursues.

(3) The authors claimed they investigated the ectosome because many proteins were altered in the TZ gene mutants. How to define ectosome related proteins? What was the significance of claiming the enrichment?

(4) Analysis of the DUF1619 domain at TCTN1 needs to be revised or removed, because the current result cannot provide any additional information. Initially, authors started this study by identifying this gene from the mutagenesis study, having the in-frame insertion (needs to be clarified whether it is in-frame insertion though) of this gene, so it would be evident that (a) deletion of this domain would make the gene non-functional, and (b) the regions outside the domain are not sufficient to rescue the gene function. If the "domain only" construct was functional, it would be informative, but it was not what the authors showed.

(5) Authors claimed the different shortening dynamics of the *tctn1* mutant, but it was not clear how much they were significant (Fig 1h). The speed looks similar, reaching zero because it is shorter.

Reviewer #4 (Remarks to the Author):

Review of Wang et al, Nature Comm.

The principal value of this report by Wang et al is the isolation of another mutant in the Transition Zone (TZ) of the model genetic organism, the biflagellate alga *Chlamydomonas*, from which most of our knowledge of the development of the cilium is derived. It is not the first mutant in this part of the ciliary apparatus, but it does, indeed, add another tool for those in the community investigating this part of the structure. It is not surprising that they find, using this mutant (*tctn1*), along with others previously isolated and described in other labs, ie. CEP290 and *nphp4*, that it is localized in the TZ region, near CEP290, by fluorescence microscopy. And it is also interesting, but not surprising, that the flagella which assemble in this mutant have differences from wt flagella. The seminal paper showing changes in flagellar protein composition in TZ mutants was by Craige et al working with Witman and Roenbaum over a decade ago (JCB 2010), and other papers have followed. However, the authors do

an excellent job of quantifying the proteomic differences between wt and the TZ mutant flagella in their comparative proteomic analysis of wt, CEP290, and NPHP4 flagella, perhaps better than has been done heretofore, and this adds to the value of the paper. I suspect that almost any disruption to the doorway to the flagellum will change its protein composition and function in receiving and releasing signals, and in its motility functions.

The authors feel that the principal new finding in this report is that the vesicles (ectosomes) actively released from the flagellar membrane are smaller than those released from wt, and that this shows the importance of the TZ in the outward signaling function of the flagella by ectosome release from the flagellar membrane. This is an interesting observation, but, again, not unexpected since the ciliary membrane, added by the exocytosis of vesicles to the peri-centriolar membrane have to pass through the TZ with their cargo of IFT and the IFT-associated cargo proteins, and other flagellar membrane moieties added at the base, in order to reach the flagellar membrane, and before they are released as ectosomes. Perhaps the best demonstration of this process has been by Kubo and colleagues, who are not quoted in this paper (and they should be), showing, in clever papers, that one can detach the flagella, of daughter cells within the mother-cell wall of *Chlamydomonas*, by pH shock in situ, and by doing so the daughter cells could not exit the mother cell wall until the daughter cells regenerated their flagella. They then hypothesized that the protease, secreted by the flagella, could not reach the cell wall. Although this work strongly implied that the protease secreted by the flagella to enable daughter cell release was in vesicles (flagellar ectosomes), this was subsequently shown by Wood in Rosenbaum's lab (CB, 2015): that the flagella were releasing vesicles (ectosomes) to enable dissolution of the mother cell wall. Therefore, the observation in Wang et al that membrane passing through the TZ, some of which is released as ectosomal vesicles from the flagellar membrane, is not a surprising observation, but it is the first careful quantification showing that the vesicles may be different when released from cells with mutations in TZ proteins, especially TZ protein which connect the membrane to the axoneme in the TZ region, as quite clearly shown in Wang et al. In regard to this, the authors feel the TCTN composes some part of the Y-links which connect the flagellar membrane to the core structures of the TZ. They say that the mutant has "largely attenuated" Y links, and I am not sure what they mean by this. The Y links are arranged radially around the TZ, associated with each of the outer doublet microtubules and it is very easy to have a thin section for EM of wt cells which does NOT show the links in these region, simply because the section misses them. What is needed (and lacking in this paper) is some quantification of the TEMs of mutant and wt re presence vs absence of the Y links.

Finally, although this report does a good job in quantifying differences in the proteomics of wt vs mutant flagella in TZ mutants, why haven't they done the same comparative analysis of TZ's isolated from mutant vs wt? There is a published method (Diener et al) for the isolation of TZs, so one wonders why, in light of this, the comparison between isolated mutant vs wt TZs was not done?

I think this work should be published in Nature Comm, not so much because it breaks new ground on how the TZ actually functions to regulate what enters and exits the flagellum (we know it does regulate) but because the mutant will provide a new tool for those doing high resolution CryoEM studies of this important region of the flagellar apparatus. I strongly suspect that the next major advances in the area on TZ function will be made by CryoEM observations of wt and mutant cells like the one isolated and analyzed in this report.

I also have some specific comments on the figures and writing. The paper needs a thorough treatment of the English used.

In Figure 2a: Is the TEM of the bulging flagellar stub from a cell with full grown, regenerating or shortening flagella, and how do they know?

In Figure 2b: Judging by the thin sections showing the H-structure in the core of the TZ, the section of the mutant is not taken in the same plane as the section of the wt. Some quantification is needed here. Also, in the X-section of the mutant, one can see some Y links.

In Figure 2 E,F,G: The authors would be well advised to try using the high resolution EM gold ab analysis of TZ protein localization as done in Craige et al. Not only does one get localization by this method, but you can actually see structure.

Figure 5, on the accumulation of non-ciliary protein (chloroplast, mitochondria etc) in the TZ mutant flagella is a new piece of evidence and is, perhaps, one of the more important aspects of this paper!

Figure 6: The quantification of the Ems of negatively-stained vesicles in wt vs TZ mutant cells is good.

Figure 7: The use of vesicle-induced adherence of mutant and wt cells to quantify vesicle "mating-activity" is clever, and a nice addition to the paper, and shows a clear difference between the activity of vesicles released from wt vs TZ mutant cells. Nice. In Figure 7d, on the comparison of ectosomes isolated from wt and mutant cells, it does not look as if the gel loads are comparable, and they should be. It is not enough to say they were isolated from the same amount of cells. If one is comparing protein composition via SDS gel of mutant vs wt, then the loads should be the same for each. This means doing a protein determination on vesicles isolated from mutants vs wt cells and then loading the same amount of protein.

In supplementary Figure 2a: the cross section of the tctn mutant flagella clearly shows the lack of the glycocalyx in the mutant flagella, and I don't think they mention it in the paper; it is important, especially since the vesicles, isolated from the mutant, were not as active in inducing the flagellar agglutination response as the wt vesicles ! The authors should also read the paper (last year?) by Bloodgood and Sloboda on the HMW glycoprotein on the outside of the flagellum.

In regard to the mis-regulation of IFT proteins in mutant flagella (Suppl), this is an interesting and important result which probably should be in the body of the paper if possible, in light of the information now available on the entry of IFT trains into the flagella via the TZ, first seen in TEM by Geimer et al, and more recently by Engel with Cryo EM in milled samples of the Chlamydomons centriole-TZ region.

One final note; One wonders why this detailed paper was submitted as a "note" or Communication when it should be submitted to a journal like JCB or JCS which will publish data-packed papers like this.

REVIEWER COMMENTS

Reviewer #1 (Remarks to the Author):

Authors Wang et al propose in their manuscript “Ciliary transition zone proteins coordinate ciliary protein composition and ectosome shedding” a role for the transition zone (TZ) protein TCTN1. In this review, I comment primarily on the mass spectrometry proteomics used. This work in part uses mass spectrometry proteomics to identify candidate proteins that are differentially abundant in WT vs tctn1 cilia. They then validate the mass spectrometry findings using immunofluorescence and immunoblotting experiments. Then, further mass spectrometry experiments are used for cilia purified from WT, tctn1, cep290, and nphp4 cells. These experiments were used to compare the differential protein abundances across the groups.

The authors have written a concise, easy to read research article. I found the rationale of their experiments clearly communicated and easy to follow, and I also appreciate that the interpretation of their results was concise and to the point. Overall, it made the manuscript easy to read and I look forward to its publication in Nature Communications as a broadly interesting investigation of the TZ protein TCTN1 and its role in ciliary formation and ectosome shedding.

Response:

We thank the reviewer for appreciating our work and helping us to improve our manuscript.

I have just a couple minor suggestions to improve the repeatability and clarity of the work, which are detailed below. - Please deposit the RAW mass spectrometry data in a repository, e.g. MASSIVE, PRIDE, etc.

Response:

Thank you for the constructive suggestion. The raw data have been deposited to the ProteomeXchange Consortium via the Proteomics Identification Database (PRIDE) partner repository with the dataset identifier PXD033090. The information and statement have also been included in the revised manuscript.

- Figures 4, 5, 6, and S6 all use “fold change” as a measure of abundance, however there is no detail how this was calculated in the Methods. Please detail how the fold change was calculated (e.g. any data normalizations or imputations) in the Methods section and label these axes and/or captions with the fold change details (e.g. log2?) so that the figures are more clear.

Response:

Thank you for the comments and suggestions.

In Fig. 4b, the volcano plot was constructed using $-\log_{10}(P \text{ value})$ against $\log_2(\text{Fold change value of } tctn1/\text{WT})$. The red and blue dots represent the up-regulated and down-regulated proteins in cilia of the *tctn1* mutant. The nonaxial vertical lines indicate the fold change of *tctn1*/WT <0.5 and >2 . The nonaxial horizontal line represents a *P* value of 0.05.

We revised the data presentation in Fig. 4c. In the revised Fig. 4c, the fold change was calculated by dividing the abundance value of each sample by the average value of three WT samples.

In Fig. 4e, the relative protein level was determined by the band intensity in western blots. First, we obtained the relative protein abundance by dividing the intensity of each protein band by the intensity of the corresponding α -tubulin band. Second, the data were normalized using the relative abundance of the corresponding WT sample, so the fold change was calculated by dividing the relative level of each sample by the relative level of the corresponding WT sample.

For the heatmap plots in the main and supplementary figures, the abundance value of each sample was listed in a matrix, in which each row represented a protein and each column represented a set of samples. We performed scaling

by dividing each value by the average abundance value of each protein in a row. Next, the scaled data were logarithmically transformed with a base of 2. The \log_2 (scaled abundance) values were used to draw the heatmap. Blue–white–red color schemes were used in the heatmap plots to visualize the individual protein expression levels in the cilia of WT and mutant cells.

We have revised the labels in the corresponding graphs and legend per your helpful suggestions.

Reviewer #2 (Remarks to the Author):

Here Wang and colleagues performed genetic screening to identify genes related to ciliogenesis using Chlamydomonas reinhardtii as a model system and reported the role of tctn1 at the ciliary transition zone, together with previously characterized cep290 and nphp4 at the same location, for transition gating to control ciliary proteome composition. Although tctn1 was reported for its association to human ciliopathy Meckel and Joubert syndrome over ten years ago (Garcia-Gonzalo, et al., 2011), we still do not fully understand how the proteins at the transition zone regulate ciliary compositions. This manuscript claimed that tctn1 has distinctive functions in transition gating compared to two known complexes (MKS complex and NPHP complex) and is also involved in determining the ciliary ectosome composition. In my opinion, proteomics analysis is critical for authors to claim the distinctive role of tctn1 compared to cep290 and nphp4, mainly showing the different ciliary proteome compositions in each condition. However, the authors did not present all information correctly, so I cannot follow this conclusion based on the results presented here.

[Major issues]

(1) All heatmap plots were not clear what they were presented. The authors described it as "relative fold-changes" but did not mention "the reference." Is it based on the protein abundance of wild-type samples (I doubt it, because wild-type samples also have varied signals)? Or the mean of all samples (how can I interpret this "fold-change")? Without proper reference, I cannot interpret any of these data.

Response:

Sorry for not clearly describing the method.

To construct the heatmap plots, the abundance value of each sample was listed in a matrix, in which each row represented a protein and each column represented a set of samples. We performed scaling by dividing each value by the average abundance value of each protein in a row. Next, the scaled data were logarithmically transformed with a base of 2. The \log_2 (scaled abundance) values were used to draw the heatmap. The details of the algorithm are shown below in **Response Fig. 1**.

Response Fig. 1. The algorithm used for the heatmaps.

We apologize that the algorithm used was not clearly described. In the heatmap plots, “the reference” for each protein is the average value (red) of the eight samples of a protein.

We have revised the methods section to describe the algorithm.

(2) In SFig6 (I think it is "the comprehensive raw data for the main figures"), almost all proteins seem to be quite different in all conditions. Not sure about cep290 and nphp4 mutants, but at least tctn1 mutants do not have that severe defect on cilia formation (after complete deciliation, the cilia is generally recovered only with shorter length). So I expected most ciliary proteomes to be preserved even in the tctn1 mutant, but this figure did not look like that. The authors need to clarify this (unless authors claimed that all > 2,000 observed proteins were significantly changed across three mutants).

Response:

The reviewer is correct. Sup Fig. 6 is the comprehensive raw data. As the reviewer noted, most ciliary proteins are present in the mutant cilia. That is why the major structure of cilia is preserved. However, as the heatmap showed, the abundances of many proteins changed, although most of the ciliary proteome was preserved.

(3) Fig 4c showed the change of fold change in cilia between WT and tct1 mutant, but again it is not clear how the authors calculated them. Like Fig 4b and Fig 4e, if the authors compared it to the wild-type cilia proteome, green should be zero, but it does not look like that.

Response:

We are sorry for not clearly describing the method.

As the reviewer pointed out, the comparison to the wild-type ciliary proteome will be a better way to present data. We revised the data presentation of Fig. 4c. In the revised Fig. 4c, the fold change was calculated by dividing the abundance value of each sample by the average value of three WT samples.

Also, based on Fig 5 and Fig 6, I guess that the authors have duplicate data. If true, these values should be presented individually rather than the mean and the error bars.

Response:

Thank you. In the revised version, we performed new proteomic experiments with triplicate samples to replace the dataset in the revised Fig. 4b and 4c.

(4) Because the authors also claimed the ectosome shedding, it would be critical to control the background/contamination of ciliary proteome. I think it is necessary to analyze the proteome in the media or similar samples to rule out the proteins not available in the ciliary structure.

Response:

Thank you for the question.

We are not sure we correctly understand the reviewer's question. In this study, we used an established cilia isolation method that has been broadly used for proteomic analysis of *Chlamydomonas* cilia (J Proteome Res. 2011, PMID: 21663328; J Cell Biol. 2005, PMID: 15998802). The purity of cilia obtained with this method has been confirmed by optical microscopy or TEM examination in previous studies.

A previous study by William Dentler (PMID: 23359798) showed that "steady-state flagella appeared to shed a minimum of 16% of their surface membrane per hour." In our study, before the excision of cilia/flagella from cell bodies, the cells were washed with fresh medium to avoid possible contamination. It took approximately 2 minutes to induce cilia excision from the time of washing. According to the ectosome shedding rate, ~0.53% of the ciliary membrane was released to the medium in the form of ectosomes within this time. Additionally, ciliary membrane proteins account for a very small proportion of the whole ciliary proteome. Moreover, ciliary ectosomes are small membrane vesicles that can only be precipitated by ultrahigh-speed centrifugation (~100,000 × g). To purify cilia,

we performed $\sim 10,000 \times g$ centrifugation, which is not sufficient to precipitate ciliary ectosomes. Considering the very low steady shedding rate of ciliary ectosomes and the very high centrifugation speed needed to precipitate them, it seems very unlikely that the ectosomes contribute much to the total protein after purification.

We also performed an experiment to evaluate the possibility of contamination. In the experiment, *Chlamydomonas* cells treated with/without cilia excision induction were used for the subsequent purification of cilia. As shown in **Response Fig. 2**, without cilia excision, no protein was detected by silver staining (**Lane 1**).

Response Fig. 2. Image of SDS-PAGE visualized by silver staining.

Lane 1. Purified ciliary fraction without cilia excision treatment. Relative loading amount, 1 ×. **Lane 2.** Loading buffer only. **Lane 3-7.** Purified ciliary fraction with cilia excision treatment. Relative loading amount, 1 × (Lane 3), 0.2 × (Lane 4), 0.1 × (Lane 5), 0.05 × (Lane 6), and 0.025 × (Lane 7).

[Minor issues]

(1) There is no description of how the authors quantify the protein abundances. The authors only described "Total peptide was used for normalization, and both unique peptides and razor peptides were used for further quantification" in the method. Is it based on spectral counting or ion intensity? Did the authors use the internal spike-in? More details about proteomic analysis should be provided.

Response:

We are sorry for not clearly describing the method.

For quantification, both unique peptides and razor peptides were used for further quantification based on the ion intensity and we did not use internal spike-in. We have also provided a detailed description of the method for the quantification in the revised manuscript.

We revised the Method sections: "Label-free quantification was performed by the summed abundances of the top 3 peptides. Both of unique peptides and razor peptides were used for label-free quantification. Peptide abundance was based on intensity of corresponding precursor ion. Quantitative protein data were further normalized by total peptide amount."

(2) The RAW data for proteomics were not deposited to the repository, like the PRIDE database. I believe that it is also essential for the reproducibility and transparency that this journal pursues.

Response:

Thank you for the constructive suggestion. The mass spectrometry proteomics data have been deposited to the

ProteomeXchange Consortium via the PRIDE partner repository with the dataset identifier PXD033090. This information has also been included in the revised manuscript.

(3) The authors claimed they investigated the ectosome because many proteins were altered in the TZ gene mutants. How to define ectosome related proteins? What was the significance of claiming the enrichment?

Response:

Thank you for the questions. Recent work with *Chlamydomonas* and *C. elegans* demonstrated that biologically active ectosomes can be released from the cilium and mediates cell–cell communications. A remarkable work (PMID: 27866888) published in Current Biology determined the protein composition of ciliary ectosomes by mass spectrometry. In this manuscript, we defined ectosome-related proteins in cilia based on the results from that work. The significance of the changes in the enrichment of ectosome-related proteins in cilia is that the changes may affect the formation and function of ectosomes, which was confirmed by our studies in Fig. 6 and Fig. 7.

(4) Analysis of the DUF1619 domain at TCTN1 needs to be revised or removed, because the current result cannot provide any additional information. Initially, authors started this study by identifying this gene from the mutagenesis study, having the in-frame insertion (needs to be clarified whether it is in-frame insertion though) of this gene, so it would be evident that (a) deletion of this domain would make the gene non-functional, and (b) the regions outside the domain are not sufficient to rescue the gene function. If the "domain only" construct was functional, it would be informative, but it was not what the authors showed.

Response:

Thank you for the kind suggestion. The authors agree with the reviewer that the information in this part is very limited, and we have removed this section in the revised manuscript.

(5) Authors claimed the different shortening dynamics of the tctn1 mutant, but it was not clear how much they were significant (Fig 1h). The speed looks similar, reaching zero because it is shorter.

Response:

Thank you for the comments. We included the average shortening rates of the cell lines in the main text for clarity. The results are shown as ciliary disassembly kinetics ($\sim 5.864 \mu\text{m/h}$ in *tctn1* and $\sim 3.363 \mu\text{m/h}$ in WT) corresponding to Fig. 1h.

Reviewer #4 (Remarks to the Author):

Review of Wang et al, Nature Comm.

*The principal value of this report by Wang et al is the isolation of another mutant in the Transition Zone (TZ) of the model genetic organism, the biflagellate alga Chlamydomonas, from which most of our knowledge of the development of the cilium is derived. It is not the first mutant in this part of the ciliary apparatus, but it does, indeed, add another tool for those in the community investigating this part of the structure. It is not surprising that they find, using this mutant (*tctn1*), along with others previously isolated and described in other labs, ie. CEP290 and *nphp4*, that it is localized in the TZ region, near CEP290, by fluorescence microscopy. And it is also interesting, but not surprising, that the flagella which assemble in this mutant have differences from wt flagella. The seminal paper showing changes in flagellar protein composition in TZ mutants was by Craig et al working with Witman and Roenbaum over a decade ago (JCB 2010), and other papers have followed. However, the authors do an excellent job of quantifying the*

proteomic differences between wt and the TZ mutant flagella in their comparative proteomic analysis of wt, CEP290, and NPHP4 flagella, perhaps better than has been done heretofore, and this adds to the value of the paper. I suspect that almost any disruption to the doorway to the flagellum will change its protein composition and function in receiving and releasing signals, and in its motility functions.

The authors feel that the principal new finding in this report is that the vesicles (ectosomes) actively released from the flagellar membrane are smaller than those released from wt, and that this shows the importance of the TZ in the outward signaling function of the flagella by ectosome release from the flagellar membrane. This is an interesting observation, but, again, not unexpected since the ciliary membrane, added by the exocytosis of vesicles to the pericentriolar membrane have to pass through the TZ with their cargo of IFT and the IFT-associated cargo proteins, and other flagellar membrane moieties added at the base, in order to reach the flagellar membrane, and before they are released as ectosomes.

Perhaps the best demonstration of this process has been by Kubo and colleagues, who are not quoted in this paper (and they should be) ,

Response:

Thank you for the suggestion. The reference has been cited in the revised version.

showing, in clever papers, that one can detach the flagella, of daughter cells within the mother-cell wall of Chlamydomonas, by pH shock in situ, and by doing so the daughter cells could not exit the mother cell wall until the daughter cells regenerated their flagella. They then hypothesized that the protease, secreted by the flagella, could not reach the cell wall. Although this work strongly implied that the protease secreted by the flagella to enable daughter cell release was in vesicles (flagellar ectosomes), this was subsequently shown by Wood in Rosenbaum's lab (CB, 2015): that the flagella were releasing vesicles (ectosomes) to enable dissolution of the mother cell wall. Therefore, the observation in Wang et al that membrane passing through the TZ, some of which is released as ectosomal vesicles from the flagellar membrane, is not a surprising observation, but it is the first careful quantification showing that the vesicles may be different when released from cells with mutations in TZ proteins, especially TZ protein which connect the membrane to the axoneme in the TZ region, as quite clearly shown in Wang et al. In regard to this, the authors feel the TCTN composes some part of the Y-links which connect the flagellar membrane to the core structures of the TZ. They say that the mutant has "largely attenuated" Y links, and I am not sure what they mean by this. The Y links are arranged radially around the TZ, associated with each of the outer doublet microtubules and it is very easy to have a thin section for EM of wt cells which does NOT show the links in these region, simply because the section misses them. What is needed (and lacking in this paper) is some quantification of the TEMs of mutant and wt re presence vs absence of the Y links.

Response:

Thank you for the suggestion. In the revised manuscript, we have quantified the presence of Y-links in WT and the *tctn1* mutant to avoid the missing of Y-links caused by EM techniques. The results are shown in Fig. 2c.

Finally, although this report does a good job in quantifying differences in the proteomics of wt vs mutant flagella in TZ mutants, why haven't they done the same comparative analysis of TZ's isolated from mutant vs wt? There is a published method (Diener et al) for the isolation of TZs, so one wonders why, in light of this, the comparison between isolated mutant vs wt TZs was not done?

Response:

Thank you for the suggestion. The structural and compositional comparison of the TZ between mutant vs. WT will be interesting. In the current study, we focused on the molecular function of the ciliopathy-related protein TCTN1

and the regulation of the ciliary composition and ectosomes by TZ proteins. Limited by the scope of this manuscript, we did not isolate TZ for comparison. We believe that systematically comparing the ultrastructure by TOMO-cryo-EM and the composition by proteomics analysis will provide new information about the transition zone and ciliopathies in future studies.

I think this work should be published in Nature Comm, not so much because it breaks new ground on how the TZ actually functions to regulate what enters and exits the flagellum (we know it does regulate) but because the mutant will provide a new tool for those doing high resolution CryoEM studies of this important region of the flagellar apparatus. I strongly suspect that the next major advances in the area on TZ function will be made by CryoEM observations of wt and mutant cells like the one isolated and analyzed in this report.

Response:

We thank the reviewer for the appreciation of our work.

I also have some specific comments on the figures and writing. The paper needs a thorough treatment of the English used.

Response:

Thank you for the suggestion. We are sorry for the writing and we revised our grammar with the help from American Journal Experts, affiliated with Springer Nature.

In Figure 2a: Is the TEM of the bulging flagellar stub from a cell with full grown, regenerating or shortening flagella, and how do they know?

Response:

Thank you for the question. In Fig. 1e, we found that cilia of the *tctn1* mutant occasionally became bulges, so we carried out TEM to observe the ultrastructure of the bulges. In Fig. 2a, mutant cells hatching from the mother cell wall for 10-15 minutes were harvested. According to the data shown in Fig. 1g and 1f, we speculate that cells hatching within 10-15 minutes will theoretically be assembling their cilia. In support of this speculation, bulges on cilia were not easily found after the cells had finished ciliary assembly, 3 hours after hatching.

In Figure 2b: Judging by the thin sections showing the H-structure in the core of the TZ, the section of the mutant is not taken in the same plane as the section of the wt. Some quantification is needed here. Also, in the X-section of the mutant, one can see some Y links.

Response:

Thank you for the comments. We used a correct section regarding the H-structure. Compared with WT cells, the *tctn1* mutant showed a disrupted Wedge-shaped structure at the transition zone. We quantified the sections (Fig. 2b). Meanwhile, we also revised the manuscript and displayed the same cross section of the TZ to show the variations in WT and the mutant (Fig. 2c, left panel) and quantified the ratio of the presence of Y-links in WT and the *tctn1* mutant (Fig. 2c, right panel).

In Figure 2 E,F,G: The authors would be well advised to try using the high resolution EM gold ab analysis of TZ protein localization as done in Craige et al. Not only does one get localization by this method, but you can actually see structure.

Response:

Thank you for the suggestion. We performed TEM analysis with gold-labeled antibodies with glutaraldehyde or high-

pressure freezer fixation. Unfortunately, the antibody labeling did not work after glutaraldehyde fixation in our hands. Using the high-pressure freezer fixation, we observed that TCTN1 was mainly localized at the transition zone periphery and associated with remnants of the transition zone membrane, although the TZ structures were not well preserved due to our relatively unsophisticated immunogold labeling skills. This result was presented in Fig. 3e. Further, we also performed super-resolution imaging to confirm the relative position of TCTN1, CEP290, and NPHP4 (Supplementary Fig. 2).

Figure 5, on the accumulation of non-ciliary protein (chloroplast, mitochondria etc) in the TZ mutant flagella is a new piece of evidence and is, perhaps, one of the more important aspects of this paper!

Response:

Thank you for your appreciation.

Figure 6: The quantification of the Ems of negatively-stained vesicles in wt vs TZ mutant cells is good.

Response:

Thank you for the kind comments.

Figure 7: The use of vesicle-induced adherence of mutant and wt cells to quantify vesicle “mating-activity” is clever, and a nice addition to the paper, and shows a clear difference between the activity of vesicles released from wt vs TZ mutant cells. Nice.

Response:

Thank you for your appreciation.

In Figure 7d, on the comparison of ectosomes isolated from wt and mutant cells, it does not look as if the gel loads are comparable, and they should be. It is not enough to say they were isolated from the same amount of cells. If one is comparing protein composition via SDS gel of mutant vs wt, then the loads should be the same for each. This means doing a protein determination on vesicles isolated from mutants vs wt cells and then loading the same amount of protein.

Response:

Thank you for the suggestion.

In Fig. 7d, we run the SDS-PAGE by loading the ciliary ectosome samples purified from equal number of mating gametes to elucidate that the total protein amount of ciliary ectosomes varies due to the loss of function of different TZ genes, thus indicating that the total bioactivity of the ciliary ectosomes is affected.

Per your suggestion, we have run SDS-PAGE with equal protein loading to detect the proteins changed among distinct TZ mutants. The new data are shown in Supplementary Figure 7.

In supplementary Figure 2a: the cross section of the tctn mutant flagella clearly shows the lack of the glycocalyx in the mutant flagella, and I don't think they mention it in the paper; it is important, especially since the vesicles, isolated from the mutant, were not as active in inducing the flagellar agglutination response as the wt vesicles ! The authors should also read the paper (last year?) by Bloodgood and Sloboda on the HMW glycoprotein on the outside of the flagellum.

Response:

Thank you for the nice comments. We did miss this. We acquired more TEM images and confirmed that the

glycocalyx in the *tctn1* flagella was disrupted. The results are shown in the revised manuscript as Fig. 2e and 2f.

In regard to the mis-regulation of IFT proteins in mutant flagella (Suppl), this is an interesting and important result which probably should be in the body of the paper if possible, in light of the information now available on the entry of IFT trains into the flagella via the TZ, first seen in TEM by Geimer et al, and more recently by Engel with Cryo EM in milled samples of the Chlamydomons centriole-TZ region.

Response:

Thank you for the constructive suggestion. IFT proteins are key regulators of the transportation of proteins into the cilia and responsible for most ciliary functions. Due to limited space in this figure, only one representative IFT protein was selected and is shown in the main figure, other similar results of IFT proteins are included in the supplementary figures.

One final note; One wonders why this detailed paper was submitted as a “note” or Communication when it should be submitted to a journal like JCB or JCS which will publish data-packed papers like this.

Response:

Thank you for your appreciation of our work.

REVIEWERS' COMMENTS

Reviewer #1 (Remarks to the Author):

The authors have addressed my previous suggestions, namely through the addition of more detailed text regarding the proteomics methods and analyses. I note that, while data seems to have been deposited on PRIDE repository, the project is still private and I would request that the authors make this data available publicly.

Reviewer #2 (Remarks to the Author):

The authors addressed most of the issues that the other three reviewers and I raised and polished the text and figure legends more clearly in the revised manuscript. So I recommend it to be published in Nature Communication in principle. One last suggestion is the scale of heatmaps in Fig 5 and Fig 6. Because the positive signal in red and the negative signal in blue are all presented on different scales, I think it can mislead the dynamics of protein abundance in each mutant. It will be more accurate if the color scales are balanced or even converted to standard statistics like Z-score.

Reviewer #4 (Remarks to the Author):

My concerns re the ms on TZ mutants and composition of the ectosomes have been addressed.

REVIEWER COMMENTS

Reviewer #1 (Remarks to the Author):

The authors have addressed my previous suggestions, namely through the addition of more detailed text regarding the proteomics methods and analyses. I note that, while data seems to have been deposited on PRIDE repository, the project is still private and I would request that the authors make this data available publicly.

Response:

We thank the reviewer for appreciating our work and helping us to improve our manuscript.

As the reviewer pointed out, our proteomic data have been deposited into the PRIDE repository. Unfortunately, we tried to publish the dataset but failed, because the PRIDE website required a PubMedID or DOI, without the information of which the request to make it public will fail.

As soon as the manuscript is assigned with a PubMedID or DOI, we are able to publish the raw data.

Reviewer #2 (Remarks to the Author):

The authors addressed most of the issues that the other three reviewers and I raised and polished the text and figure legends more clearly in the revised manuscript. So I recommend it to be published in Nature Communication in principle.

Response:

We thank the reviewer for appreciating our work.

One last suggestion is the scale of heatmaps in Fig 5 and Fig 6. Because the positive signal in red and the negative signal in blue are all presented on different scales, I think it can mislead the dynamics of protein abundance in each mutant. It will be more accurate if the color scales are balanced or even converted to standard statistics like Z-score.

Response:

We thank the reviewer for the constructive suggestion. The heatmaps in Fig. 5 and 6 have been revised by normalizing the data with Z-score. We have also revised the Method section and the Figure legend section.

Reviewer #4 (Remarks to the Author):

My concerns re the ms on TZ mutants and composition of the ectosomes have been addressed.

Response:

We thank the reviewer for appreciating our work and helping us to improve our manuscript.